# Widespread mermithid nematode parasitism of Cretaceous insects

Cihang Luo[1,2]*, George O Poinar[3], Chunpeng Xu[1,2], De Zhuo[4], Edmund A Jarzembowski[1], Bo Wang[1]*

[1]State Key Laboratory of Palaeobiology and Stratigraphy, Nanjing Institute of Geology and Palaeontology and Center for Excellence in Life and Paleoenvironment, Chinese Academy of Sciences, Nanjing, China; [2]University of Chinese Academy of Sciences, Beijing, China; [3]Department of Integrative Biology, Oregon State University, Corvallis, United States; [4]Beijing Xiachong Amber Museum, Beijing, China

**Abstract** Mermithid nematodes are obligate invertebrate parasites dating back to the Early Cretaceous. Their fossil record is sparse, especially before the Cenozoic, thus little is known about their early host associations. This study reports 16 new mermithids associated with their insect hosts from mid-Cretaceous Kachin amber, 12 of which include previously unknown hosts. These fossils indicate that mermithid parasitism of invertebrates was already widespread and played an important role in the mid-Cretaceous terrestrial ecosystem. Remarkably, three hosts (bristletails, barklice, and perforissid planthoppers) were previously unknown to be parasitized by mermithids both past and present. Furthermore, our study shows that in contrast to their Cenozoic counterparts, Cretaceous nematodes including mermithids are more abundant in non-holometabolous insects. This result suggests that nematodes had not completely exploited the dominant Holometabola as their hosts until the Cenozoic. This study reveals what appears to be a vanished history of nematodes that parasitized Cretaceous insects.

*For correspondence:
chluo@nigpas.ac.cn (CL);
bowang@nigpas.ac.cn (BW)

**Competing interest:** The authors declare that no competing interests exist.

## Editor's evaluation

This important study greatly expands our knowledge of the fossil record of mermithid nematodes, modern members of which are ecologically important parasitoids of arthropods, annelids and mollusks today. The most important finding is that mermithids parasitized a number of insect clades in the Cretaceous that they are not known to infect today or in Cenozoic amber. The evidence for a shift in exploited hosts from non-holometabolous insects in the mid-Cretaceous to holometabolous ones by the Eocene is exceptionally well supported by statistical analysis; potential collection bias is addressed as well and ruled out.

## Introduction

Nematodes (roundworms), a group of non-segmented worm-like invertebrates, are some of the most abundant animals on earth in terms of individuals (*Lorenzen, 1994*; *Poinar, 2011*). They are distributed worldwide in almost all habitats and play key roles in ecosystems by linking soil food webs, influencing plant growth and facilitating nutrient cycling (*Yeates et al., 2009*; *van den Hoogen et al., 2019*; *Zhang et al., 2020*). The earliest known definite nematode fossil occurs in the Lower Devonian Rhynie Chert inside the cortex cells of an early land plant and has been considered to be a plant parasite (*Poinar et al., 2008*), but unconfirmed nematode-like fossils and trace fossils may date back to the Precambrian (*Poinar, 1979*; *Parry et al., 2017*; *De Baets et al.,*

*2021a*). Despite their abundance in many extant ecosystems, nematodes are exceedingly rare in the fossil record, since most of them are small, with soft bodies and concealed habits (*Poinar, 2011*).

The Mermithidae represent a family of nematodes that are obligate invertebrate parasites which occur in insects, millipedes, crustaceans, spiders, molluscs, and earthworms (*Nickle, 1972*; *Poinar, 1979*). They can affect the morphology, physiology, and even the behavior of their hosts (*Petersen, 1985*). The life cycle of mermithids comprises five stages (*Poinar and Otieno, 1974*; *Poinar, 1983*; *Poinar, 2001b*). Eggs are deposited in the environment, and the developing embryos moult once and emerge from the eggs as second-stage juveniles. These juveniles are the infective stage that enter the hemocoel of potential hosts. After a relatively rapid growth phase (third stage), mermithids exit hosts as postparasitic juveniles (fourth stage). They then enter a quiescent phase and moult twice to become adults (*Poinar, 2015a*). They tend to exit their hosts even before maturation, though, if their hosts are stressed (*Poinar, 2015b*). Hosts usually die when the mermithids exit, which is why mermithids have been widely studied as possible biological control agents, especially against aquatic stages of medically important insects like mosquito larvae (*Petersen, 1985*). Although mermithids kill their hosts like parasitoids, they are commonly considered as parasites like other nematodes (*De Baets et al., 2021a*; *De Baets et al., 2021b*).

Palaeontological records of parasitism provide critical clues about the origination and diversification of important nematode groups, and elucidate the synecology and coevolution of ancient parasites and their hosts. Due to their relatively large size and invertebrate-parasitic habits, mermithid nematodes are most likely of all nematodes to occur as recognizable fossils, especially in amber as they exited their invertebrate hosts that became entrapped in resin (*Poinar, 2011*). Their fossil record dates back to the Early Cretaceous with *Cretacimermis libani* from a chironomid midge (Diptera: Chironomidae) in Lebanese amber (~135 Ma; million years ago) (*Poinar et al., 1994*). However, there is a dearth of examples of other Cretaceous mermithids prior to the present study, with only *Cretacimermis chironomae*, also from chironomid hosts, *Cretacimermis protus* from biting midges (Diptera: Ceratopogonidae) and *Cretacimermis aphidophilus* from an aphid (Hemiptera: Burmitaphididae) in mid-Cretaceous Kachin amber (*Poinar and Buckley, 2006*; *Poinar and Sarto i Monteys, 2008*; *Poinar, 2011*; *Poinar, 2017*).

Here, we report 16 additional mermithid nematodes associated with their insect hosts in mid-Cretaceous Kachin amber (approximately 99 million years old). These examples triple the diversity of Cretaceous mermithids (from 4 to 13 species) and reveal previously unknown host–parasite relationships. Our study also shows that mermithids were widely distributed in a number of diverse insect lineages by the mid-Cretaceous, but they preferred to parasitize non-holometabolous rather than holometabolous insects, thus providing novel insights into the early evolution of nematode parasitism of insects.

## Results

### Systematic palaeontology

Fossil nematodes can be attributed to the family Mermithidae based mainly on their relatively large size, coiled posture, and morphological comparison with extant mermithids (body shape, length, diameter, tail structure, etc.) (*Poinar, 2011*). They can also be distinguished from nematomorphs due to the lack of small elevations of irregular areas (areoles) on their epicuticles (*Poinar, 2001b*). Due to palaeontological inability to adequately detect all biological adult characters, it is impossible to place in or refer fossil mermithid nematodes to any natural extant genus. This is why fossil collective genera have been erected under the same guidelines as recent collective genera for difficult nematodes. The importance of placing nematode species in collective genera is to underpin or establish the time, place and hosts of these parasitic lineages. For Cretaceous Mermithidae not assignable to any previously known genus or lacking biologically preferred diagnostic characters, the collective genus *Cretacimermis* was erected (but invalidly, see remarks for *Cretacimermis* below) (*Poinar, 2001b*). Putative hosts were determined by noting nematodes emerging from their bodies or completely emerged nematodes adjacent to potential hosts, especially if there is physical evidence that a particular insect was parasitized.

## Family Mermithidae Braun, 1883

### Collective genus *Cretacimermis* Poinar, gen. nov.

urn:lsid:zoobank.org:act:152E262D-5A65-4EB8-A059-D380D53D32F8

**Etymology.** The generic name is derived from the combination of the prefix, 'cretac-' meaning chalky referring to the Cretaceous age of the collective, and 'Mermis' is the name of the type genus of Mermithidae. Gender: feminine.

**Included species.** *Cretacimermis adelphe* Luo & Poinar, sp. nov.; *Cretacimermis aphidophilus **Poinar, 2017***; *Cretacimermis calypta* Luo & Poinar, sp. nov.; *Cretacimermis cecidomyiae* Luo & Poinar, sp. nov.; *Cretacimermis chironomae **Poinar, 2011***; *Cretacimermis cimicis* Luo & Poinar, sp. nov.; *Cretacimermis directa* Luo & Poinar, sp. nov.; *Cretacimermis incredibilis* Luo & Poinar, sp. nov.; *Cretacimermis libani* (***Poinar et al., 1994***) ***Poinar, 2001a*** (=*Heleidomermis libani **Poinar et al., 1994***); *Cretacimermis longa* Luo & Poinar, sp. nov.; *Cretacimermis manicapsoci* Luo & Poinar, sp. nov.; *Cretacimermis protus **Poinar and Buckley, 2006***; *Cretacimermis psoci* Luo & Poinar, sp. nov.

**Diagnosis.** Mermithid nematodes not assignable to any previously known genus or lacking biologically preferred diagnostic characters from the Cretaceous. Size large compared with host, length commonly more than 5.0 mm, with length/width ratio more than 50; coiled for at least one loop; cuticle smooth and lacking cross fibres; trophosome clear but sometimes fractured; head and tail rounded or pointed.

**Age and occurrence.** Cretaceous; Lebanese and Kachin ambers.

**Remarks.** The genus name '*Cretacimermis*' was invalidly established in ***Poinar, 2001b*** due to the lack of a formal definition (***International Commission on Zoological Nomenclature, 1999***: *Art. 13.1.1*). Here, we formally erect this genus.

### *Cretacimermis incredibilis* Luo & Poinar, sp. nov. (Figures 1A and 2A–D)

urn:lsid:zoobank.org:act:22E141CC-8BEE-42F6-85BB-84243D13C1A9

**Etymology.** The species epithet is from the Latin 'incredibilis' = incredible.

**Type host.** Bristletail (Archaeognatha).

**Material.** Holotype. Kachin amber, rectangular piece, 18×7×4 mm, weight 0.4 g, specimen No. NIGP201872.

**Diagnosis.** Mermithid nematode parasitizing Archaeognatha from mid-Cretaceous Kachin amber.

**Description.** Body brownish, partially transparent (***Figure 2A***); cuticle smooth, lacking cross fibres but with fine ridges in areas of body bends (***Figure 2B***); trophosome evident (***Figure 2A***); head rounded (***Figure 2C***), tail appendage not observed (***Figure 2D***); length 17.9 mm; greatest width 80 μm; a (length/width)=224.

**Remarks.** While the nematode has completely exited from the host, the tail end is adjacent to an exit wound on the host (***Figure 2D***) indicating a true parasitic association.

### *Cretacimermis calypta* Luo & Poinar, sp. nov. (Figures 1B and 2E-H)

urn:lsid:zoobank.org:act:3F29A451-1993-4EE7-B24E-81D8A1F5B8B7

**Etymology.** The species epithet is derived from the Greek 'kalyptê' = hidden.

**Type host.** A specimen of *Burmaphlebia reifi* Bechly & Poinar (Odonata: Epiophlebioptera: Epiophlebioidea: Burmaphlebiidae).

**Material.** Holotype. Kachin amber, rhomboid piece, 19×13×5 mm, weight 1.4 g, specimen no. NIGP201870.

**Diagnosis.** Mermithid nematode parasitizing Odonata from mid-Cretaceous Kachin amber.

**Description.** Body white with clear partially transparent portions (***Figure 2E***); cuticle lacking cross fibres (***Figure 2F***); trophosome fractured (***Figure 2E***); head round (***Figure 2G***), tail bluntly rounded (***Figure 2H***); length 40.9 mm; greatest width 324 μm; a=126.

**Remarks.** Although no exit wound can be clearly found, most of the body coils are adjacent to the head of the adjacent damselfly, indicating that the nematode was just emerging from the host.

### *Cretacimermis adelphe* Luo & Poinar, sp. nov. (Figures 1C and 2I-K)

urn:lsid:zoobank.org:act:B2858585-3B4C-4E06-B5AA-7CCD37F7CC5A

**Etymology.** The species epithet is derived from the Greek 'adelphê' = sister.

**Type host.** Earwig (Dermaptera).

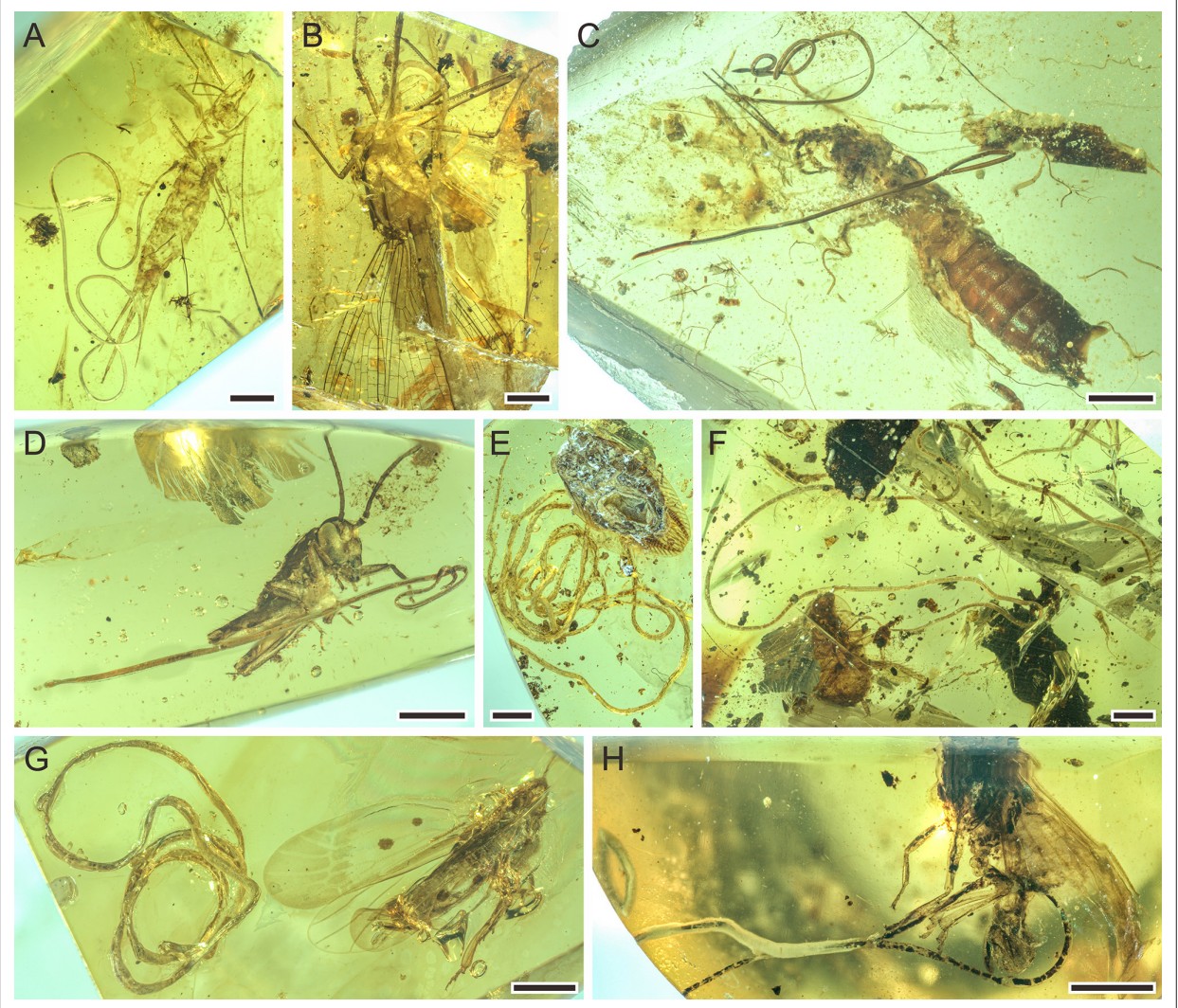

**Figure 1.** Mermithids and their insect hosts from mid-Cretaceous Kachin amber (~99 Ma; million years ago). (**A**) *Cretacimermis incredibilis* sp. nov. (holotype) adjacent to its bristletail host. (**B**) *Cretacimermis calypta* sp. nov. (holotype) adjacent to its damselfly host. (**C**) Two separate specimens of *Cretacimermis adelphe* sp. nov. (upper specimen is holotype) that have emerged from their earwig host. (**D**) *Cretacimermis directa* sp. nov. (holotype) adjacent to its cricket host. (**E**) *Cretacimermis longa* sp. nov. (holotype) adjacent to its adult cockroach host. (**F**) *Cretacimermis longa* sp. nov. (paratype) adjacent to its juvenile cockroach host. (**G**) *Cretacimermis perforissi* sp. nov. (holotype) adjacent to its perforissid planthopper host. (**H**) *Cretacimeris perforissi* sp. nov. (paratype) adjacent to second perforissid planthopper. Scale bars = 2.0 mm (**B, E, F**), 1.0 mm (**A, C, D, G**), 0.5 mm (**H**).

**Material.** Kachin amber, cabochon, 16×13×3 mm, weight 0.4 g, specimen no. NIGP201876.

**Diagnosis.** Mermithid nematode parasitizing Dermaptera from mid-Cretaceous Kachin amber.

**Description.** Upper specimen (holotype): body dark gray, opaque, coiled (*Figure 2I*); cuticle smooth, lacking cross fibres; head narrowed with acute tip (*Figure 2J*), tail blunt (*Figure 2K*); length 7.4 mm, greatest width 67 µm, a=110. Lower specimen (paratype): body dark gray, opaque, outstretched; head point-blunted, length at least 7.5 mm, greatest width 76 µm.

**Remarks.** The posterior part of the abdomen of the earwig has been damaged, so it is most likely that these mermithids exited from the host through this wound.

*Cretacimermis directa* Luo & Poinar, sp. nov. (Figures 1D, 3A and B)

urn:lsid:zoobank.org:act:3565ACFE-BD87-441D-9923-DC75F640EDC2

**Etymology.** The species epithet is from the Latin 'directa'=arranged in a straight line.

**Type host.** An early instar cricket nymph (Orthoptera: Ensifera: Grylloidea).

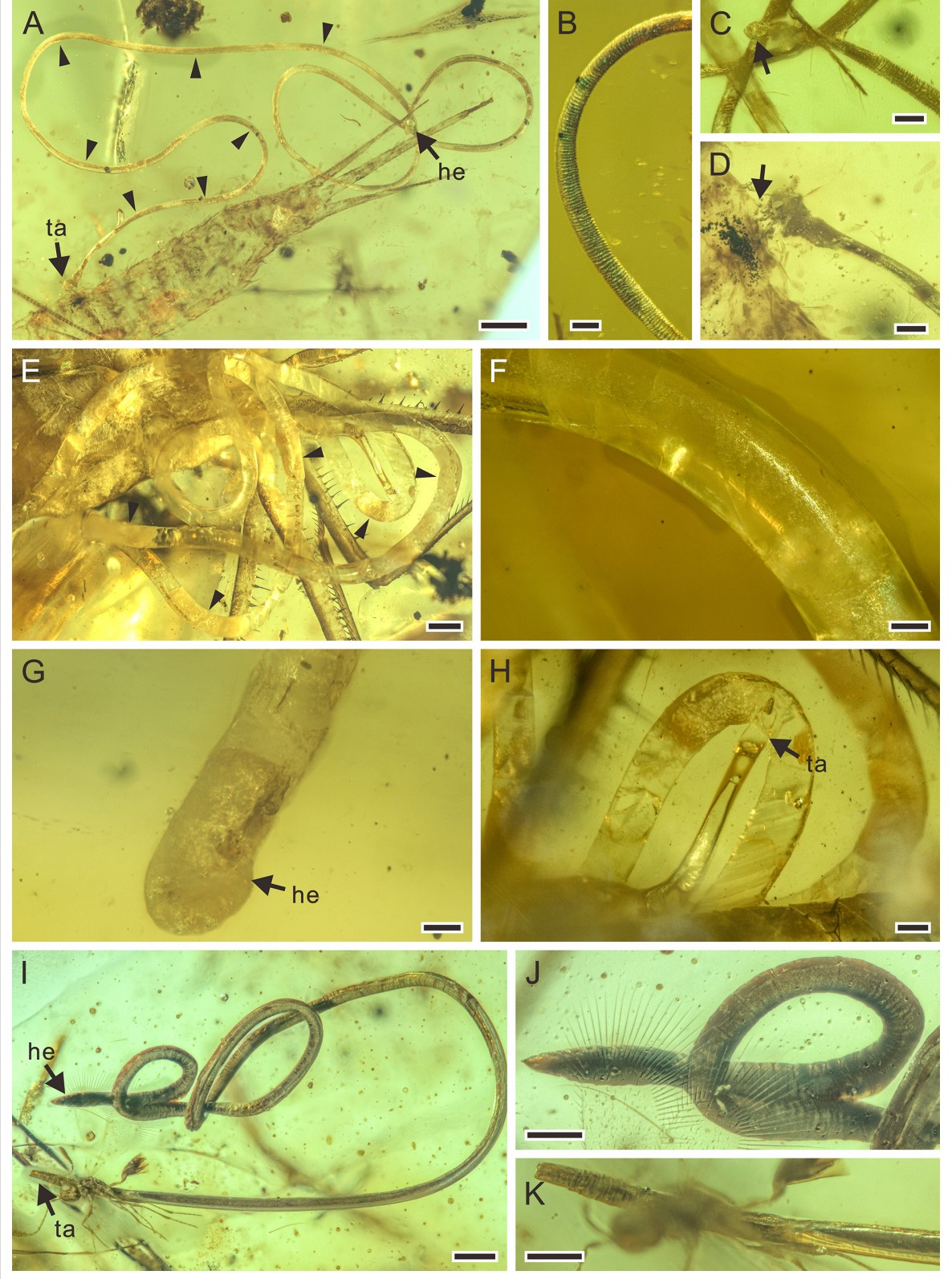

**Figure 2.** Detailed photographs of *Cretacimermis incredibilis* sp. nov., holotype, NIGP201872 (**A–D**), *Cretacimermis calypta* sp. nov., holotype, NIGP201870 (**E–H**), and *Cretacimermis adelphe* sp. nov. (upper specimen is holotype and lower specimen is paratype), NIGP201876 (**I–K**). (**A**) Habitus of *C. incredibilis*, some trophosome remains are marked by triangular black arrows. (**B**) Fine ridges in areas of body bends. (**C**) Head (arrowed). (**D**) Tail and the exit wound on the host (arrowed). (**E**) Habitus except head part of *C. calypta*, some trophosome remains are marked by triangular black arrows.

*Figure 2 continued on next page*

*Figure 2 continued*

(**F**) Detail of body. (**G**) Head. (**H**) Tail. (**I**) Habitus of upper specimen (holotype), opaque body and pointed head. (**J**) Detail of head. (**K**) Detail of tail. Scale bars = 0.5 mm (**A, E**), 0.2 mm (**H, I**), 0.1 mm (**B–D, F, G, J, K**). Abbreviations: he, head; ta, tail.

**Material.** Holotype. Kachin amber, cabochon, 18.5×4.5×3 mm, weight 0.3 g, specimen no. NIGP201873.

**Diagnosis.** Mermithid nematode parasitizing Orthoptera from mid-Cretaceous Kachin amber.

**Description.** Body well preserved, elongate except for a small coil at anterior end, grayish; trophosome slightly fractured in a few areas; head rounded (*Figure 3A*), tail bluntly pointed (*Figure 3B*); length 8.8 mm; greatest width 98 μm; a=90.

**Remarks.** There is no distinct wound on this cricket's body, but the nematode is adjacent to it, and there is no other insect nearby. Therefore, it is most likely that the mermithid had just emerged from the host.

### *Cretacimermis longa* Luo & Poinar, sp. nov. (Figures 1E,F and 3C-J)

urn:lsid:zoobank.org:act:4EE8DDA3-FEB8-4B4E-ABD2-2D9A9587FB1A

**Etymology.** The species epithet is from the Latin 'longa'=long.

**Type host.** Cockroaches of the extinct Family Mesoblattinidae (Blattodea) (*Figure 3C and G*).

**Material.** Kachin amber. First piece (holotype): cabochon, 20×13×2 mm, weight 0.5 g, specimen no. NIGP201875; second piece (paratype): cabochon, 35×24×6 mm, weight 5.2 g, specimen no. NIGP201877.

**Diagnosis.** Mermithid nematode parasitizing mesoblattinid cockroach from mid-Cretaceous Kachin amber.

**Description.** Nematode from adult cockroach, first piece (*Figures 1E and 3C–F*): body light gray, speckled, partially transparent; head narrow, tail obscured; length at least 104.3 mm, greatest width 310 μm. Nematode from juvenile cockroach, second piece (*Figures 1F and 3G–J*): body light to dark gray; head rounded, tail obscured; length at least 59.9 mm; greatest width 246 μm.

**Remarks.** These two nematodes are still in the process of exiting. Also, the cavity in the abdomen of the adult cockroach (*Figure 3C*) indicates the location of the developing parasite.

### *Cretacimermis perforissi* Luo & Poinar, sp. nov. (Figures 1G,H and 4)

urn:lsid:zoobank.org:act:32C1C1E1-C177-4E25-90FE-E5D48A0E471E

**Etymology.** The species epithet is derived from the type genus of Perforissidae.

**Type host.** Planthopper of the extinct Family Perforissidae (Hemiptera: Fulgoromorpha).

**Material.** Kachin amber. First piece (holotype): trapezoid, 9×7×3 mm, weight 0.2 g, specimen no. NIGP201868; second piece (paratype): semicircular, 8×4×4 mm, weight 0.1 g, specimen no. NIGP201878.

**Diagnosis.** Mermithid nematode parasitizing perforissid planthopper from mid-Cretaceous Kachin amber.

**Description.** First piece (*Figure 4A–E*): body complete, mostly grayish and opaque, coiled several times; trophosome evident in some body areas; cuticle smooth, lacking cross fibres but with fine ridges in areas of body bends; head blunt; tail pointed; length 22.2 mm; greatest width 183 μm; a=121. Second piece (*Figure 4F–H*): body incomplete with single coil, mostly black and opaque; trophosome fractured; cuticle smooth; head and tail missing; total length unknown; greatest width 109 μm.

**Remarks.** The abdomen of the first perforissid planthopper is empty, which probably contained the developing nematode. The abdomen of the second perforissid planthopper is broken, which is consistent with an emerging mermithid.

### *Cretacimermis manicapsoci* Luo & Poinar, sp. nov. (Figures 5A,B and 6A-G)

urn:lsid:zoobank.org:act:B0C34BAB-A6F5-4E39-86CF-D87A1A61C2E4

**Etymology.** The species epithet is derived from the type genus of Manicapsocidae.

**Type host.** Barklouse of the Family Manicapsocidae (Psocodea) (*Figure 6A, E*).

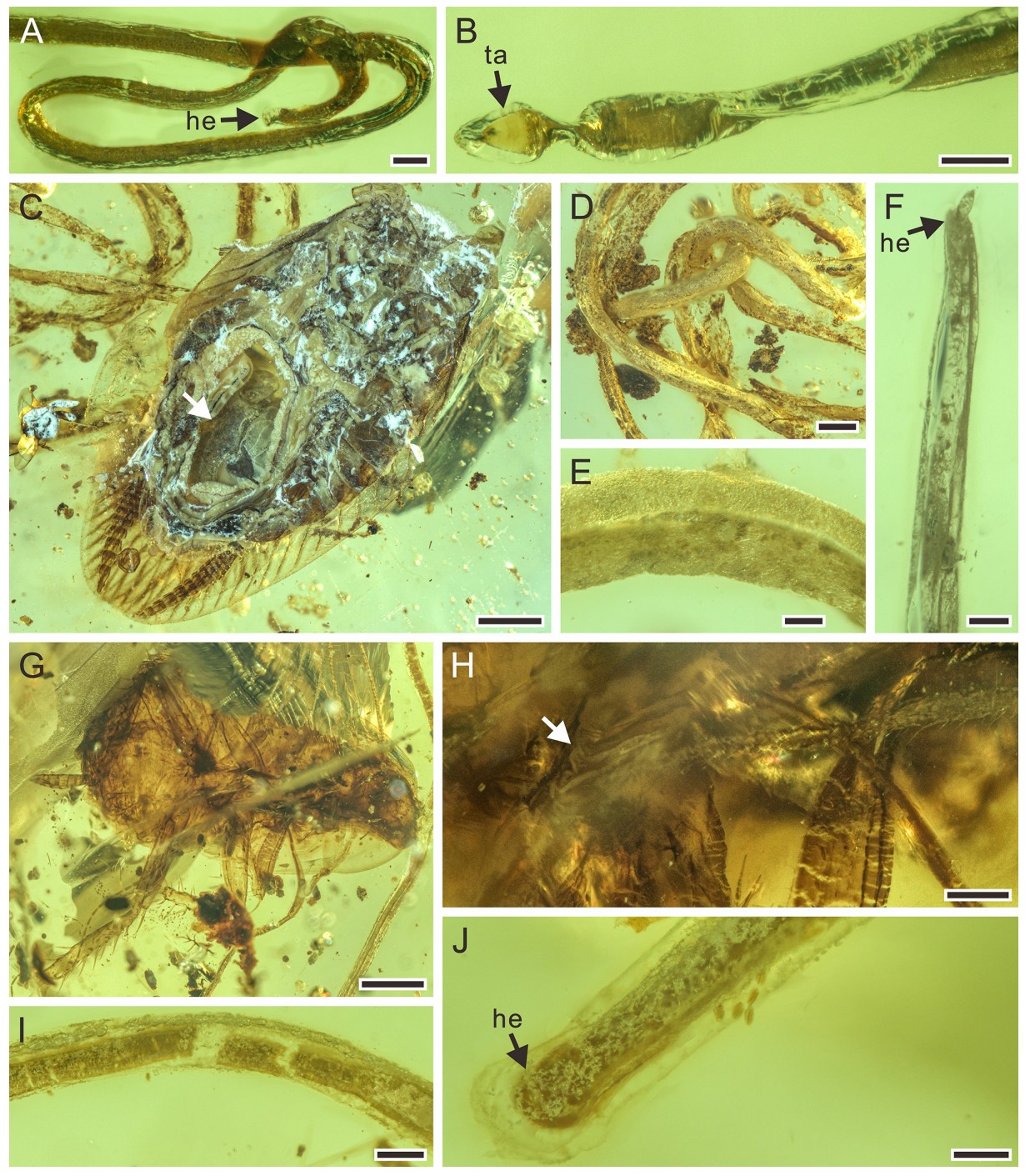

**Figure 3.** Detailed photographs of *Cretacimermis directa* sp. nov., holotype, NIGP201873 (**A, B**) and *Cretacimermis longa* sp. nov., holotype, NIGP201875 (**C–F**), paratype, NIGP201877 (**G–J**). (**A**) Detail of head. (**B**) Detail of tail. (**C**) Host, an adult of Mesoblattinidae (Blattodea), note the hollow abdomen (arrowed) that probably contained the developing nematode. (**D**) Detail of body. (**E**) Enlarged details of body. (**F**) Head. (**G**) Host, a juvenile of Mesoblattinidae (Blattodea). (**H**) The termination of the nematode, note the mermithid was in the process of emerging from the host's body (arrowed). (**I**) Detail of body. (**J**) Head, showing loose outer cuticle. Scale bars = 1.0 mm (**C, G**), 0.5 mm (**D**), 0.2 mm (**H, I**), 0.1 mm (**A, B, E, F, J**). Abbreviations: he, head; ta, tail.

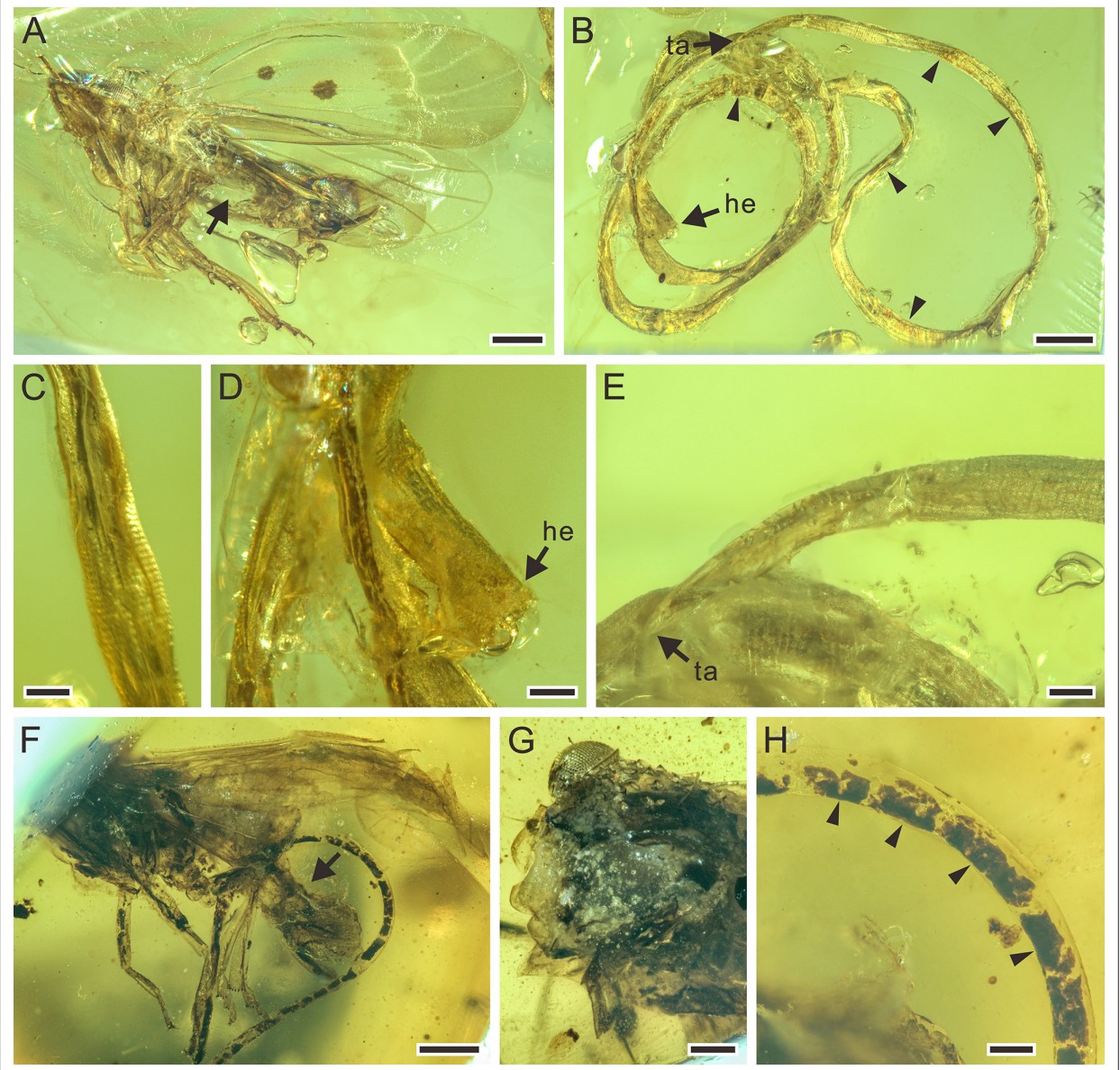

**Figure 4.** Detailed photographs of *Cretacimermis perforissi* sp. nov., holotype, NIGP201868 (**A–E**) and paratype, NIGP201878 (**F–H**). (**A**) Host, Perforissidae (Hemiptera: Fulgoromorpha), note the hollow abdomen (arrowed) which probably contained the developing nematode. (**B**) Habitus of the coiled body of *C. perforissi*, some trophosome remains are marked by triangular black arrows. (**C**) Detail of body, note artefact ridges on cuticle. (**D**) Head. (**E**) Tail. (**F**) Host, Perforissidae (Hemiptera: Fulgoromorpha), note the broken abdomen that is probably due to the emergence of the mermithid. (**G**) Front view of host, indicating it is a perforissid planthopper. (**H**) Detail of body, showing smooth cuticle and dark, fractured trophosomes (arrowed). Scale bars = 0.5 mm (**A, B, F**), 0.2 mm (**G**), 0.1 mm (**C–E, H**). Abbreviations: he, head; ta, tail.

**Material.** Kachin amber. First piece (holotype): cabochon, 34×15×3 mm, weight 1.3 g, specimen no. NIGP201879; second piece (paratype): cabochon, 13×9×2 mm, weight 0.1 g, specimen no. NIGP201880.

**Diagnosis.** Mermithid nematode parasitizing manicapsocid barklouse from mid-Cretaceous Kachin amber.

**Description.** First piece (*Figure 6A–D*): body complete, essentially a dark tube inside a clear tube, bent several times, one bend overlapping leg of host; lacking cross fibres but with fine ridges; cuticle smooth; head blunt; tail narrowed; length 34.0 mm; greatest width, 87 µm; a=391. Second piece (*Figure 6E–G*): body incomplete, uneven, mostly grayish and opaque, coiled twice; trophosome

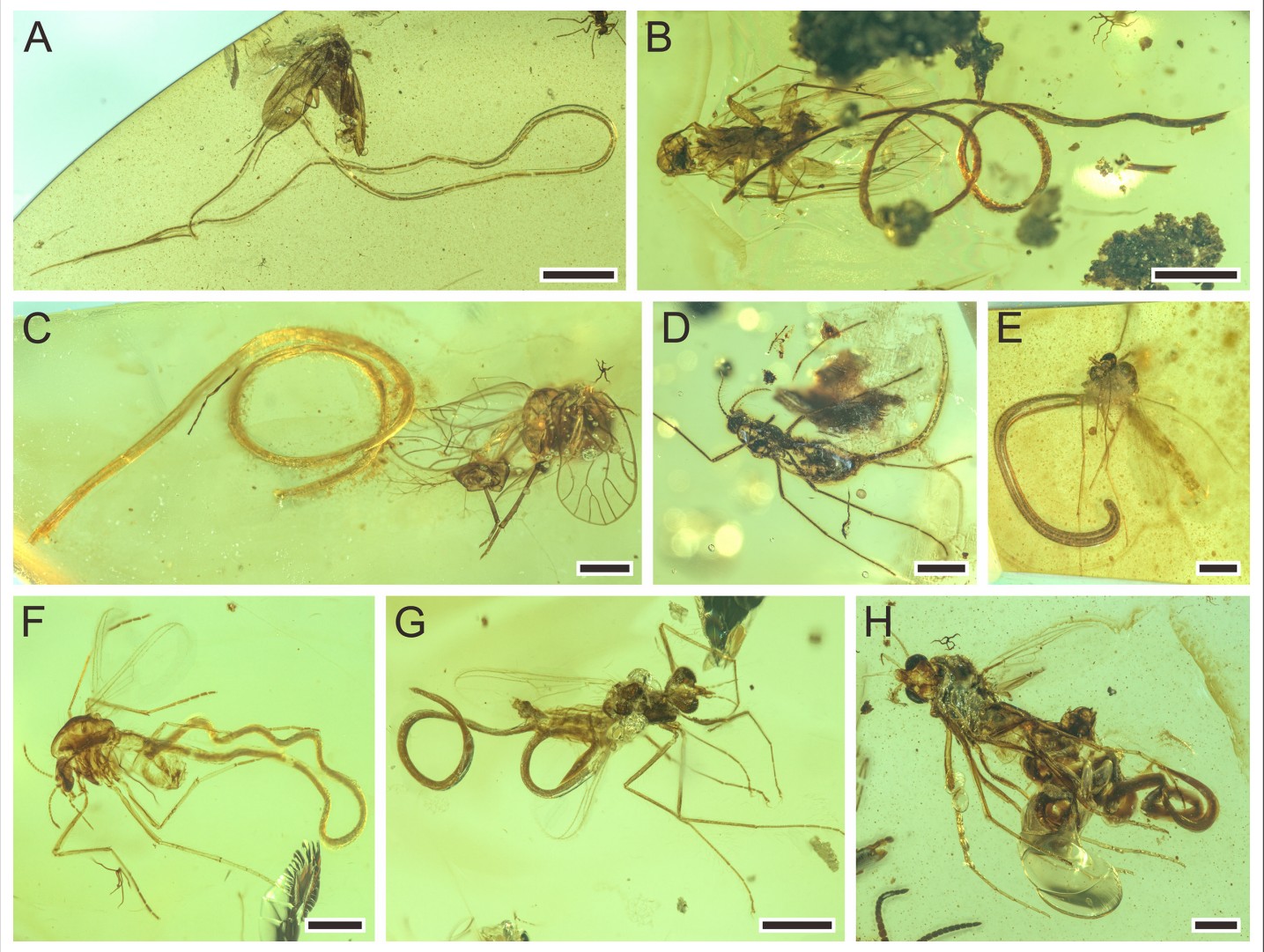

**Figure 5.** Mermithids and their insect hosts from mid-Cretaceous Kachin amber. Part II. (**A**) *Cretacimermis manicapsoci* sp. nov. (holotype) adjacent to its manicapsocid barklouse host. (**B**) *Cretacimermis manicapsoci* sp. nov. (paratype) adjacent to second manicapsocid barklouse host. (**C**) *Cretacimermis psoci* sp. nov. (holotype) adjacent to its compsocid barklouse host. (**D**) *Cretacimermis cecidomyiae* sp. nov. (holotype) emerging from its gall midge (cecidomyiid) host. (**E–H**) Four specimens of *Cretacimermis chironomae* **Poinar, 2011** emerging from their chironomid hosts. Scale bars = 2.0 mm (**A**), 1.0 mm (**B**), 0.5 mm (**C–H**).

evident, cuticle lacking cross fibres; head blunt, tail missing; length of remaining body 12.7 mm; greatest width 107 µm.

**Remarks.** First piece: there is no distinct wound on the barklouse's body, but the nematode is adjacent to it, and there is no other sizeable insect nearby. Therefore, it is most likely that the mermithid had just emerged from the host. Second piece: the abdomen of the second perforissid planthopper is partly lost, which is consistent with an emerging mermithid.

## *Cretacimermis psoci* Luo & Poinar, sp. nov. (Figures 5C and 6H-J)

urn:lsid:zoobank.org:act:13C60738-C5FA-4EBC-914A-1E68149DF282

**Etymology.** The species epithet is derived from New Latin 'psocus'=member of psocopteran lineage.

**Type host.** Barklouse of the family Compsocidae (Psocodea) (*Figure 6H*).

**Material.** Holotype. Kachin amber, cabochon, 12×4×2 mm, weight 0.1 g, specimen no. NIGP201874.

**Diagnosis.** Mermithid nematode parasitizing compsocid barklouse from mid-Cretaceous Kachin amber.

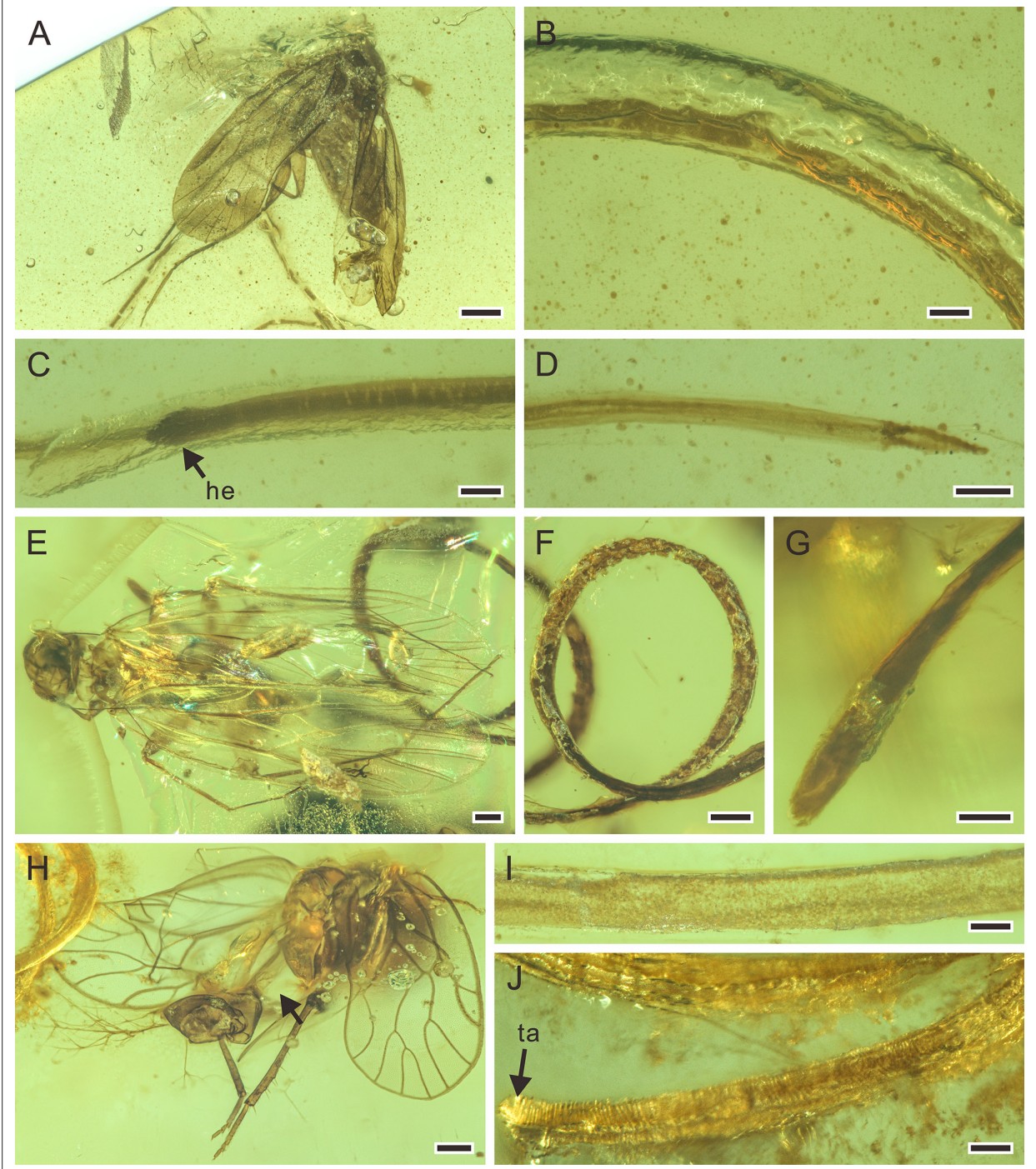

**Figure 6.** Detailed photographs of *Cretacimermis manicapsoci* sp. nov., holotype, NIGP201879 (**A–D**), paratype, NIGP201880 (**E–G**), and *Cretacimermis psoci* sp. nov., holotype, NIGP201874 (**H–J**). (**A**) Barklouse host, Manicapsocidae (Psocoptera). (**B**) Detail of body. (**C**) Head. (**D**) Tail. (**E**) Barklouse host, Manicapsocidae. (**F**) Coiled body. (**G**) Head. (**H**) Barklouse host, Compsocidae (Psocoptera), note the broken abdomen that is probably due to the emergence of the mermithid. (**I**) Detail of body. (**J**) Tail, note artefactual cuticular ridges. Scale bars = 0.5 mm (**A**), 0.2 mm (**E, F, H**), 0.1 mm (**B–D, G, I, J**). Abbreviations: he, head; ta, tail.

**Description.** Body incomplete, tanned, partially transparent (*Figure 6I*); cuticle with areas of shrinkage ridges; head missing, tail rounded (*Figure 6J*); length 9.6 mm; greatest width 111 µm; a=87.

**Remarks.** The nematode is adjacent to the host and the empty abdomen indicates the area that contained the developing parasite.

*Cretacimermis cecidomyiae* Luo & Poinar, sp. nov. (Figures 5D and 7A-C)

urn:lsid:zoobank.org:act:507A7843-3FC1-4366-9268-833D751E236B

**Etymology.** The species epithet is derived from the type genus of Cecidomyiidae.

**Type host.** Gall midge of the family Cecidomyiidae (Diptera: Culicomorpha) (*Figure 7A*).

**Material.** Holotype. Kachin amber, cabochon, 7×4×3 mm, weight 0.1 g, specimen no. NIGP201871.

**Diagnosis.** Mermithid nematode parasitizing cecidomyiid midge from mid-Cretaceous Kachin amber.

**Description.** Body grayish with white areas; cuticle smooth, lacking cross fibres; body with dark trophosome; head missing, tail obscured; length at least 1.5 mm; greatest width 73 μm; a=at least 21 (*Figure 7B and C*).

**Remarks.** The mermithid is preserved in the process of emerging from the host's body.

*Cretacimermis chironomae* Poinar, 2011 (Figures 5E-H and 7D-Q)

**Type host.** A non-biting midge of the family Chironomidae (Diptera: Culicomorpha).

**Material.** Kachin amber. First piece: subtrapezoidal, 9.5×3.5×2 mm, weight 0.1 g, specimen no. NIGP201869; second piece: cabochon, 14×9×4 mm, weight 0.5 g, specimen no. LYD-MD-NG001; third piece: cabochon, 16×13×4 mm, weight 0.6 g, specimen no. LYD-MD-NG002; fourth piece: cabochon, 18×10×2 mm, weight 0.1 g, specimen no. NIGP201881.

**Description.** First piece (*Figure 7D–H*): body grayish, uniform with little distortion; cuticle smooth, lacking cross fibres; details of trophosome clear; head rounded; length at least 3.6 mm, greatest width 193 μm. Second piece (*Figure 7I–K*): body well preserved, with distinct opaque trophosome; length at least 4.9 mm, greatest width 106 μm. Third piece (*Figure 7L–N*): body well preserved, mostly opaque due to trophosome; posterior body portion flattened and slightly twisted; head pointed; length at least 4.3 mm, greatest width 75 μm. Fourth piece (*Figure 7O–Q*): both specimens' body dark brown, very wide in relation to length; cuticle smooth, lacking cross fibres; trophosome opaque; heads rounded, tails obscured; lengths not possible to attain, greatest widths 175 μm and 155 μm, respectively.

**Remarks.** These mermithids were in the process of emerging from the hosts' bodies.

## Discussion

The sixteen new mermithids associated with their insect hosts described above include 10 insect–mermithid associations. The hosts of nine species were previously unrecorded, which triples the diversity of Cretaceous Mermithidae (from 4 to 13 species). In today's ecosystems, mermithids have been reported from a variety of arthropods, in a range of environments, and often infecting large percentages of host populations and causing mass mortality (*Poinar, 1975*; *Poinar, 1979*; *Petersen, 1985*). Despite their abundance in extant terrestrial ecosystems, mermithids are rare in the fossil record as they are not readily preserved as fossils. Twenty-two fossil mermithid species have been described from the Cenozoic with their hosts (*Supplementary file 1*: Table S1), mainly from Eocene Baltic amber (11 species) and Miocene Dominican amber (9 species), but only four pre-Cenozoic species associated with only two insect orders have previously been recorded (*Poinar and Buckley, 2006*; *Poinar and Sarto i Monteys, 2008*; *Poinar, 2011*; *Poinar, 2017*). However, according to our new records, nine insect orders are now known to have been infested by mermithid nematodes in Kachin amber and this number is even higher than that of Baltic amber (~45 Ma) and Dominican amber (~18 Ma) (six and three insect orders, respectively), despite a much longer time spent searching for nematodes in the latter two amber deposits (*Poinar, 2011*). Together with previously described mermithids in Kachin amber (*Poinar, 2001b*; *Grimaldi et al., 2002*; *Poinar and Buckley, 2006*; *Poinar, 2011*; *Poinar, 2017*), our results suggest that mermithid parasitism of insects was actually widespread during the mid-Cretaceous (*Figure 8*). Mermithids species are usually characterized by strong host specificity, they are specific to a single species or to one or two families of insects, and are almost always lethal to their hosts (*Stoffolano, 1973*; *Petersen, 1985*), thus our study indicates that the widespread mermithid parasitism probably already played an important role in regulating the population of insects in Cretaceous terrestrial ecosystems.

Our study provides new information on fossil host–parasite associations, including three previously unknown host–mermithid associations and first fossil records of four host associations. One is *Cretacimermis incredibilis* sp. nov., which has completely exited from a bristletail (Archaeognatha). Its tail end

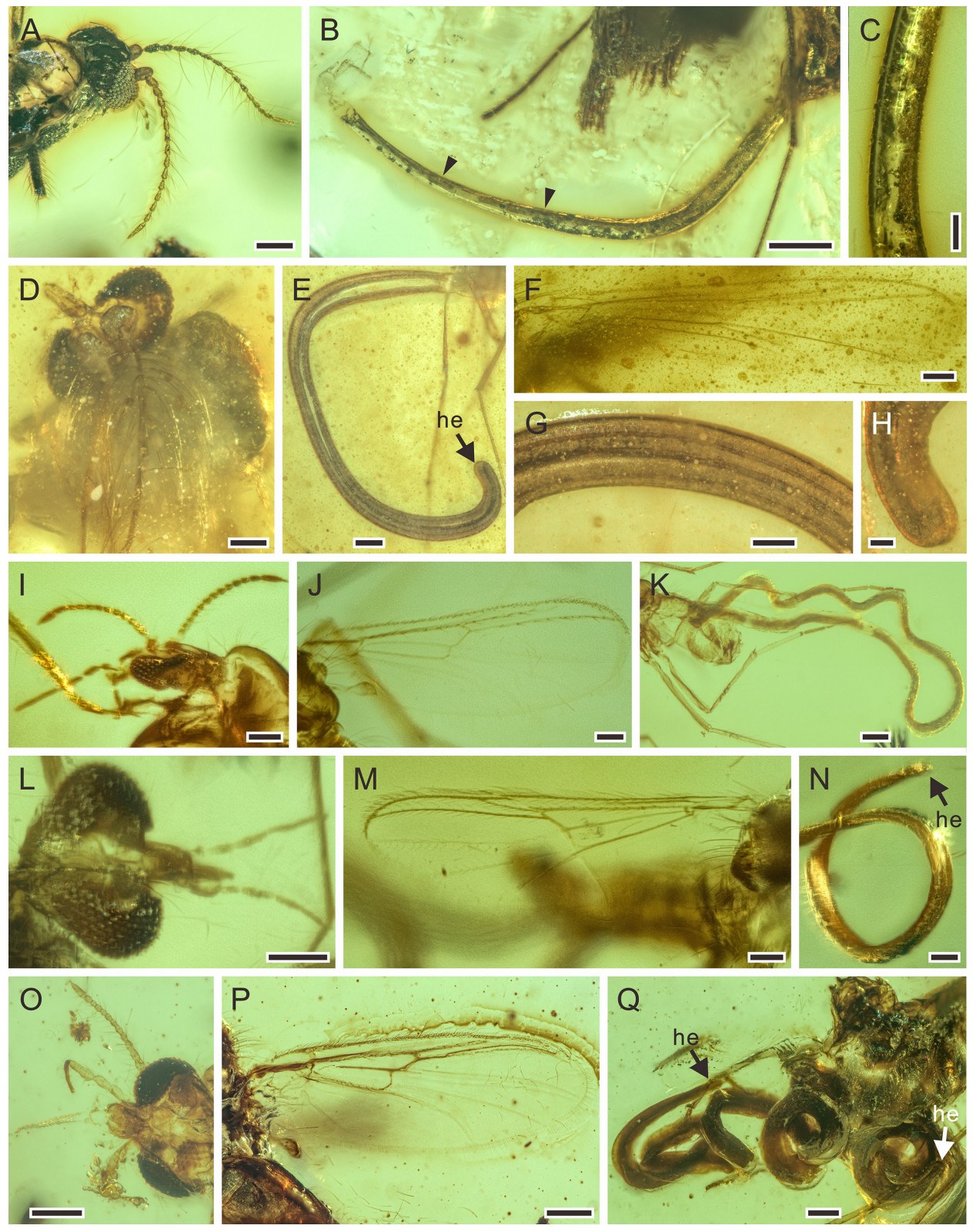

**Figure 7.** Mermithids and their Diptera hosts from mid-Cretaceous Kachin amber. (**A–C**) *Cretacimermis cecidomyiae* sp. nov., holotype, NIGP201871. (**A**) Head of cecidomyiid host. (**B**) Habitus of nematode, some trophosome remains are marked by triangular black arrows. (**C**) Detail of body. (**D–H**) First piece with *Cretacimermis chironomae* **Poinar, 2011**, NIGP201869. (**D**) Detail of head of host. (**E**) Habitus of nematode. (**F**) Forewing venation of host. (**G**) Detail of body. (**H**) Detail of head. (**I–K**) Second piece of *C. chironomae*, LYD-MD-NG001. (**I**) Detail of head of host. (**J**) Forewing venation of host.

*Figure 7 continued on next page*

*Figure 7 continued*

(**K**) Habitus of nematode, note that a portion of the mermithid is still in the host's abdomen. (**L–N**) Third piece with *C. chironomae*, LYD-MD-NG002. (**L**) Detail of head of host. (**M**) Forewing venation of host. (**N**) Detail of head and body. (**O–Q**) Fourth piece with *C. chironomae*, NIGP201881. (**O**) Detail of head of host. (**P**) Forewing venation of host. (**Q**) Habitus of two nematodes. Scale bars = 0.2 mm (**B**, **E**, **K**, **O–Q**), 0.1 mm (**A**, **D**, **F**, **G**, **I**, **J**, **L–N**), 50 μm (**C**, **H**). Abbreviation: he, head.

is still adjacent to an exit wound on the host (*Figure 2D*), indicating a true parasitic association. There are no previous extant or extinct records of nematodes attacking bristletails (*Poinar, 1975*; *Poinar, 2011*). A second new mermithid–host association is barklice (Psocodea) with three different specimens parasitized by mermithids. No barklice are parasitized by mermithids today (*Poinar, 1975*), but our specimens imply that such relationship might have been quite common in the mid-Cretaceous. Two members of the extinct planthopper family Perforissidae were also parasitized by mermithids, thus providing the oldest record of mermithid parasitism of planthoppers. The mermithid *Heydenius brownii* parasitized achiliid planthoppers in Baltic amber (*Poinar, 2001a*) and this association also occurs in extant planthoppers (*Choo et al., 1989*; *Helden, 2008*). Furthermore, our findings are the first fossil records of mermithids parasitizing dragonflies (Odonata), earwigs (Dermaptera), crickets (Orthoptera) and cockroaches (Blattodea), four host associations predicted from extant records (*Poinar, 1975*).

Nematode body fossils are scarce and mainly known from amber (*De Baets et al., 2021b*), sometimes together with their hosts (*Poinar, 2011*). To explore the evolution of nematode–host relationship, we compiled nematode–host records in the three best-studied amber biotas (mid-Cretaceous Kachin amber, Eocene Baltic amber and Miocene Dominican amber; *Supplementary file 1*: Table S1). Our results indicate that not only the mermithids, but also the nematodes as a whole, experienced a certain degree of host transition between the Cretaceous and Cenozoic (*Figure 9*). We cannot fully exclude the possibility of collection bias, but its influence is probably low because Kachin amber has been extensively studied in the last two decades and its biota has already become the most diverse known amber biota; moreover, holometabolous insects are much more diverse in the collections than non-holometabolous ones (1296 vs 465 species: *Ross, 2023*). It is therefore unlikely that holometabolous insects are underrepresented among the known hosts of mermithids. Among the insect hosts of mermithids preserved in Kachin amber, only one of the nine orders (Diptera) is holometabolous (i.e. insects with 'complete' metamorphosis), whilst it is four out of six (Hymenoptera, Trichoptera, Lepidoptera and Diptera) in Baltic amber and all three insect host orders (Hymenoptera, Coleoptera and Diptera) are holometabolous in Dominican amber. The situation is similar when referring to the amount of nematode parasitism (*Table 1*). In Kachin amber, only about 40% of the hosts (in total, not only insects) are holometabolous, while this percentage increases to 80% in Baltic and Dominican amber and this result is acceptable when uncertainty is considered (*Figure 9B*). Diptera are the most common hosts of nematodes from all three amber biotas; also, the oldest fossil animal that was found to host a mermithid is a dipteran from Early Cretaceous Lebanese amber (*Poinar et al., 1994*). This is probably because most dipteran larvae develop in moist or aquatic environments that are particularly suitable habitats for nematodes (*Poinar, 2011*). It is evident that Holometabola are the most important hosts of extant mermithids as well as all invertebrate-parasitizing nematodes (*Poinar, 1975*) and this hexapod subgroup dominated the insect fauna during the Cretaceous (*Labandeira and Sepkoski, 1993*; *Labandeira, 2005*; *Sohn et al., 2015*; *Peters et al., 2017*; *Zhang et al., 2018*; *Thomas et al., 2020*; *Wang et al., 2022*). Our study suggests that, except for Diptera, nematodes had not completely exploited Holometabola as hosts in the mid-Cretaceous. This suggests that non-holometabolous insects (i.e. insects without 'complete' metamorphosis) were more available as hosts in the mid-Cretaceous and the widespread association between nematodes and Holometabola might have formed later.

Finally, discovering these nematodes in mid-Cretaceous Kachin amber brings new opportunities to study the evolution of parasitism through the medium of amber. Amber is a unique form of fossilization (*Hsieh and Plotnick, 2020*). Although amber is patchily distributed in space and time, it is still especially suitable for investigating the evolution of terrestrial parasites associated with arthropods due to preservation potential (*De Baets and Littlewood, 2015*; *Leung, 2017*; *De Baets et al., 2021a*; *De Baets et al., 2021b*; *Leung, 2021*; *Poinar, 2021*). The high diversity of mermithid nematodes

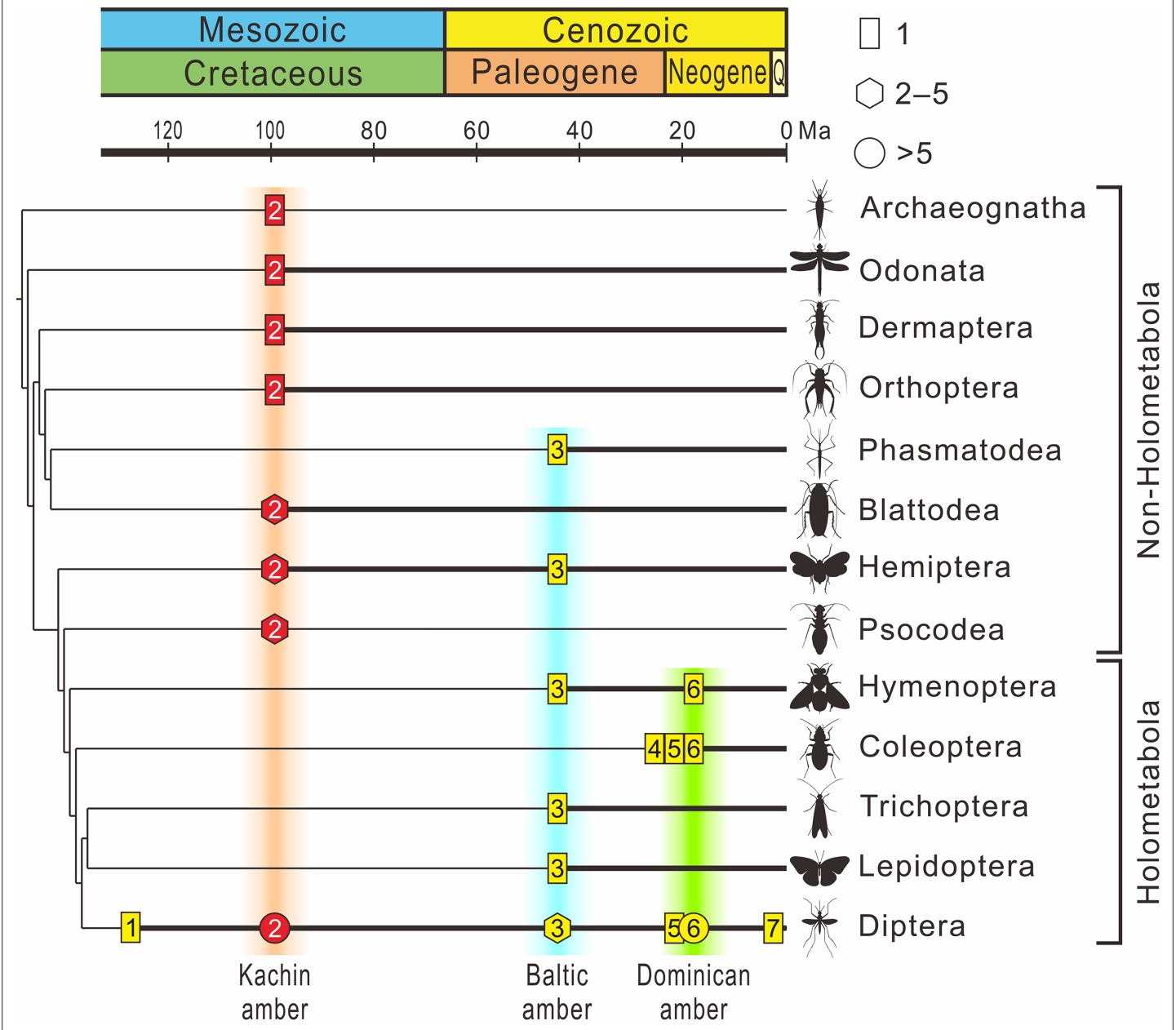

**Figure 8.** The fossil record of Mermithidae plotted on the phylogenetic tree of insects. The chronogram of the insect tree is modified from *Misof et al., 2014* (thin black line); insect orders without a fossil record of mermithid parasitism are excluded. Thick black lines indicate the presence of mermithid parasitism. Rectangles represent the fossil number of species of mermithid not exceeding one, hexagons represent the fossil number of mermithids between two and five (inclusively), and circles represent the fossil number of mermithids more than five. Yellow coloration within the symbols represents previous records, red coloration within the symbols represents records in this paper. 1 – Lebanese amber, Early Cretaceous, approximately 135 Ma; 2 – Kachin amber, mid-Cretaceous, approximately 99 Ma; 3 – Baltic amber, Eocene, approximately 45 Ma; 4 – Rhine lignite (brown coal), Oligocene/Miocene, approximately 24 Ma; 5 – Mexican amber, Early Miocene, approximately 20 Ma; 6 – Dominican amber, Miocene, approximately 18 Ma. 7 – Willershausen, Kreis Osterode, Germany, Late Pliocene, approximately 3 Ma.

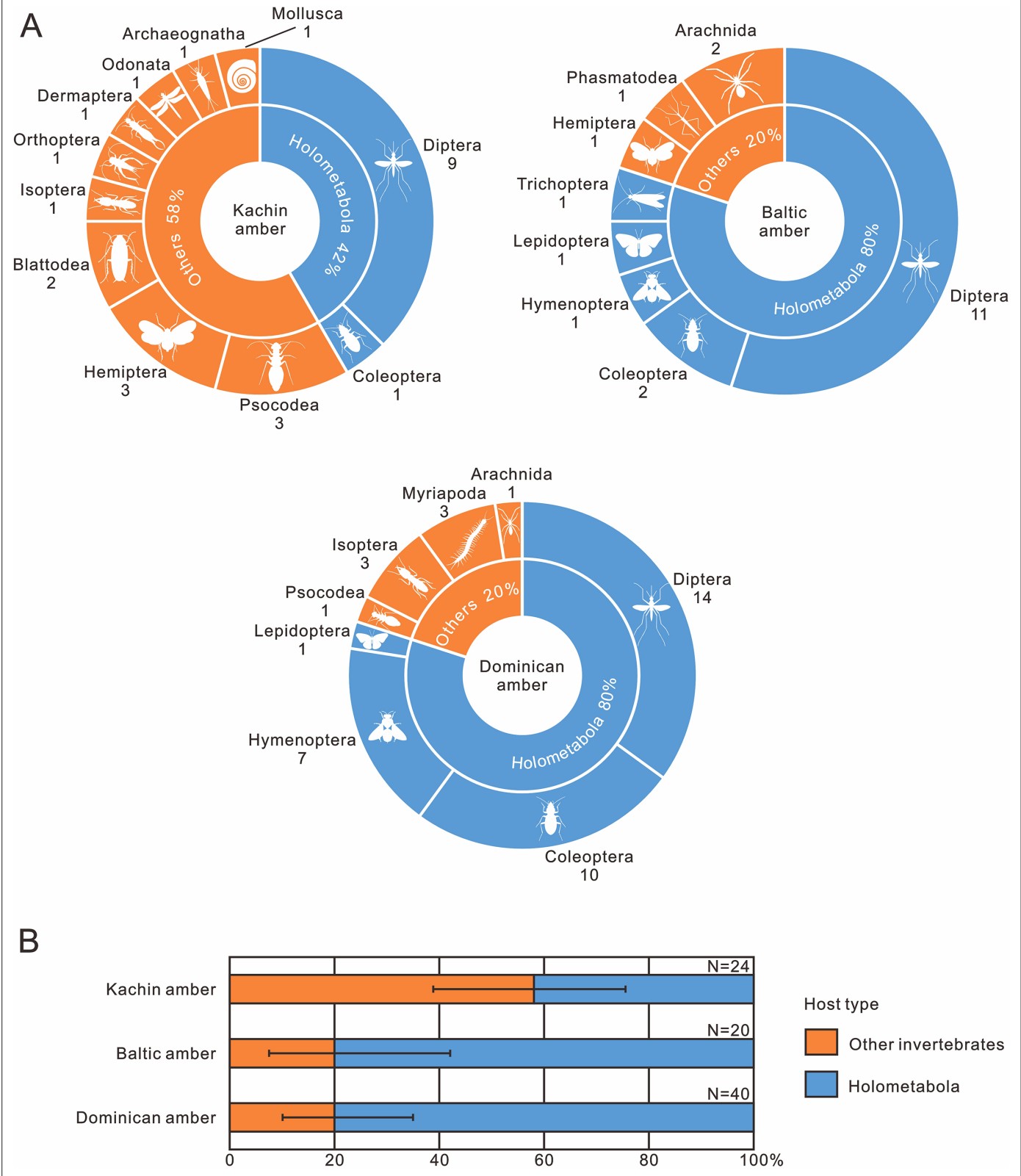

**Figure 9.** The occurrence frequency of invertebrate–nematode associations from mid-Cretaceous Kachin amber (~99 Ma), Eocene Baltic amber (~45 Ma) and Miocene Dominican amber (~18 Ma). (**A**) Pie diagrams, the quantity of fossil species is indicated below the orders. (**B**) Stacked bar plots, error bars represent 95% binomial confidence intervals (for data, see also **Table 1**).

**Table 1.** The quantity of invertebrate–nematode associations from the mid-Cretaceous Kachin amber (~99 Ma), Eocene Baltic amber (~45 Ma) and Miocene Dominican amber (~18 Ma).

| Amber source | Host type | Quantity | FOI | 95% CI |
|---|---|---|---|---|
| Kachin amber | Holometabola | 10 | 58.33% | 38.80–75.56% |
| | Other invertebrates | 14 | | |
| Baltic amber | Holometabola | 16 | 20.00% | 7.49–42.18% |
| | Other invertebrates | 4 | | |
| Dominican amber | Holometabola | 32 | 20.00% | 10.24–35.01% |
| | Other invertebrates | 8 | | |

95 % CI is calculated using the Agresti-Coull method of the "binom.confint" function from the binom *R* package (https://cran.r-project.org/package=binom) of *R* 4.2.2. Abbreviations: FOI, frequency of other invertebrates; CI, confidence intervals.

during the mid-Cretaceous as shown here provides a glimpse into the structure of ancient parasitic nematode–host associations and their evolution over the past 100 million years.

## Materials and methods
### Provenance and deposition
The specimens described here are from the Cretaceous deposits in the Hukawng Valley located southwest of Maingkhwan in Kachin State (26°20′ N, 96°36′ E) in Myanmar (*Thu and Zaw, 2017*). Radiometric U–Pb zircon dating determined the age to be 98.79±0.62 Ma (*Shi et al., 2012*), a date consistent with an ammonite trapped in the amber (*Yu et al., 2019*).

Fourteen specimens (NIGP201868–201881) are deposited in the NIGPAS, and two specimens (LYD-MD-NG001, 002) are deposited in Linyi University. The fossils were collected in full compliance with the laws of Myanmar and China (work on this manuscript began in early 2016). To avoid any confusion and misunderstanding, all authors declare that to their knowledge, the fossils reported in this study were not involved in armed conflict and ethnic strife in Myanmar, and were acquired prior to 2017. All specimens are permanently deposited in well-established, public museums, in full compliance with the International Code of Zoological Nomenclature and the Statement of the International Palaeoentomological Society (*International Commission on Zoological Nomenclature, 1999*; *Szwedo et al., 2020*).

### Optical photomicrography
Observations were performed using a Zeiss Stemi 508 microscope. The photographs were taken with a Zeiss Stereo Discovery V16 microscope system in the Nanjing Institute of Geology and Palaeontology, Chinese Academy of Sciences, and measurements were taken using Zen software. Photomicrographic composites of 10–150 individual focal planes were digitally stacked using the software HeliconFocus 6.7.1 for a better illustration of 3D structures. Photographs were adjusted using Adobe Lightroom Classic and line drawings were prepared using CorelDraw 2019 graphic software.

## Acknowledgements
We are grateful to George Perry, David Marjanović and two anonymous reviewers for valuable suggestions, which greatly improved this paper. We also thank Daran Zheng, Youning Su, Peter Vršanský, Adam Stroiński, and Art Borkent for help with identification of the hosts of nematodes. This research was supported by the National Natural Science Foundation of China (42125201), Strategic Priority Research Program of the Chinese Academy of Sciences (XDB26000000), the Second Tibetan Plateau Scientific Expedition and Research (2019QZKK0706) and the CAS President's International Fellowship Initiative (PIFI). This is a contribution to UNESCO-IUGS IGCP Project 679.

## Additional information

### Funding

| Funder | Grant reference number | Author |
|---|---|---|
| National Natural Science Foundation of China | 42125201 | Bo Wang |
| Chinese Academy of Sciences | Strategic Priority Research Program XDB26000000 | Bo Wang |
| Second Tibetan Plateau Scientific Expedition and Research | 2019QZKK0706 | Bo Wang |
| CAS President's International Fellowship Initiative | | Edmund A Jarzembowski |
| UNESCO-IUGS | IGCP Project 679 | Edmund A Jarzembowski |

The funders had no role in study design, data collection and interpretation, or the decision to submit the work for publication.

### Author contributions

Cihang Luo, Conceptualization, Validation, Investigation, Visualization, Methodology, Writing – original draft, Project administration, Writing – review and editing; George O Poinar, Investigation, Visualization, Writing – original draft, Writing – review and editing; Chunpeng Xu, Investigation, Visualization, Writing – review and editing; De Zhuo, Investigation, Visualization; Edmund A Jarzembowski, Writing – review and editing; Bo Wang, Conceptualization, Supervision, Funding acquisition, Validation, Investigation, Methodology, Writing – original draft, Project administration, Writing – review and editing

### Author ORCIDs
Cihang Luo (ID) http://orcid.org/0000-0002-4855-6185
Bo Wang (ID) http://orcid.org/0000-0002-8001-9937

### Decision letter and Author response
Decision letter https://doi.org/10.7554/eLife.86283.sa1
Author response https://doi.org/10.7554/eLife.86283.sa2

---

## Additional files

### Supplementary files
• Supplementary file 1. Table S1 The fossil record of nematodes, undescribed specimens are not included.
• MDAR checklist

### Data availability
All data are available in the main text and/or the supplementary materials.

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
