## [Editor Report]

This important study greatly expands our knowledge of the fossil record of mermithid nematodes, modern members of which are ecologically important parasitoids of arthropods, annelids and mollusks today. The most important finding is that mermithids parasitized a number of insect clades in the Cretaceous that they are not known to infect today or in Cenozoic amber. The evidence for a shift in exploited hosts from non-holometabolous insects in the mid-Cretaceous to holometabolous ones by the Eocene is exceptionally well supported by statistical analysis; potential collection bias is addressed as well and ruled out.

---

## [Decision Letter]

**Decision letter after peer review:**

Thank you for submitting your article "Widespread mermithid nematode parasitism of Cretaceous insects" for consideration by *eLife*. Your article has been reviewed by 3 peer reviewers including David Marjanović as the Reviewing Editor and Reviewer #1, and the evaluation has been overseen by George Perry as the Senior Editor.

Two of the reviewers have discussed their reviews with one another, and the Reviewing Editor has drafted this to help you prepare a revised submission.

Essential revisions:

1) The abstract mentions statistical analyses, but only pie charts are presented. The data you present in the supplementary material could easily, and should, be used for a statistical analysis to test your hypothesis of a shift toward holometabolous insects as hosts. Please also briefly address, in the text, the possibility of collection bias: are, for example, beetles underrepresented in museum collections of insects in amber, and are such biases likely to be identical for all sources of amber (Baltic amber has been sampled for centuries, Kachin amber only for a decade or two)?

2) Please write for a slightly wider audience. For example, please explain how trophosomes are recognized, and how fossil mermithids can be differentiated from, for example, nematomorphs; please also mention if mermithids are known to exit their hosts as a panic reaction. If you can illustrate and describe the exit wounds, this would greatly strengthen the manuscript against the possibility that some of the associations might be accidental.

3) The nomenclature needs a few minor improvements. I also urge you to take this opportunity to publish the genus name Cretacimermis properly, because that has not been done so far – the original publication lacks a diagnosis, which is required even for collective groups, so the name "Cretacimermis" is not currently available and does not even compete for homonymy.

*Reviewer #1 (Recommendations for the authors):*

I am not familiar with nematodes or amber, so I only wrote my review after receiving and reading the other reviews, and I have focused on the topics I am more competent in.

Comments on science

Given the wide readership of *eLife*, it would be good to explicitly state the evidence that these rather featureless elongate fossils represent specifically the parasitic stage of, specifically, mermithids (and not for example nematomorphs or a whole new group of nematodes). In your descriptions you mention trophosomes (implying parasitism) and details of the cuticle; is any of this diagnostic for Mermithidae, or is something else that you can see? How are trophosomes recognized? (I see the brown fragments you point out in Figure 4H; why do you interpret them as a trophosome?) Is it possible to illustrate or at least describe the exit wounds in more detail in the specimens where the mermithids have fully exited?

Can you exclude collection bias as a factor in the apparent shift to holometabolous hosts?

Line 25: By "statistical analyses", do you mean the pie charts in Figure 9? That's a presentation of data, not an analysis. The term made me expect mathematics.

64: Are mermithids known to exit their hosts as a flight reaction when the host is stressed?

Comments on nomenclature

I was wrong in my pre-review: collective groups are explicitly allowed under the ICZN; and diagnoses do not need to be limited to morphological characters, they can refer to behavior. However, collective species are not allowed – only collective genera and subgenera (ICZN Articles 10.3, 42.2.1). The species names you propose are therefore fine, just don't call them "collective".

However, because genus names must be published with a diagnosis even if they're intended for a collective genus, "Cretacimermis" is actually unavailable: for purposes of zoological nomenclature, this name does not exist. I quote the original publication (Poinar 2001b: 262): "Since the generic placement of this specimen was based in part on its host family, it is now prudent to transfer the species libani from Heleidomermis Rubstov into a new genus Cretacimermis Poinar. Thus, this nematode, which is the oldest definite fossil nematode, should now be called Cretacimermis libani Poinar, Acra and Acra (1994)." This is not "a description or definition that states in words characters that are purported to differentiate the taxon" (ICZN Art. 3.1.1). I therefore strongly recommend that you validly publish Cretacimermis here as a new name. (Interestingly, you don't need to select a type species, because collective groups do not have type species: ICZN Art. 13.3.2, 42.3.1, 66, 67.14. This must be precisely because species cannot be collective groups.)

I also strongly recommend against using times or places in diagnoses (if I find, say, a mermithid exiting an archaeognath in Late Cretaceous amber from New Jersey, do I really have to erect a new species name for it instead of referring it to "Cretacimermis" incredibilis…?), let alone the type of preservation (amber), but this is not forbidden by the ICZN. ("Character", as used in Art. 3.1.1, is defined in the Glossary as: "Any attribute of organisms used for recognizing, differentiating, or classifying taxa.")

Line 101: Why "extraordinary" and not simply "incredible"?

129: The species name itself is fine, but adelphos means "brother", not "sister"; "sister" is "adelphê".

144: Much simpler: "The species epithet is the Latin 'directa' = arranged in a straight line."

172, 175, 193, 225: Another reviewer has pointed out that not all bugs are bedbugs, and that naming a planthopper parasite after bedbugs may therefore not be a good idea. Given that you name "C." manicapsoci after the type genus of Manicapsocidae (not directly after Manicapsocidae) and likewise "C." cecidomyiae after they type genus of Cecidomyiidae, you could also name the planthopper parasite "C. perforissi", after the type genus of Perforissidae.

226: Being a valid name, Cecidomyia should be preceded by a comma rather than surrounded by quotation marks. You can also delete it entirely; you didn't spell out Manicapsocus in line 193.

286: Replace "Archeognatha" by "Archaeognatha".

365: Is he Peter or Petr?

593-594 and Supplementary Table: As you correctly state in the text, the authors of "C." chironomae are not Poinar, 2011, but Grimaldi et al., 2002.

Comments on style and language

Line 27-28: "have […] exploited" is from the point of view of the present, so it clashes with "until the Cenozoic"; replace it by "had […] exploited".

34: Replace "one" by "some" to fit the plural "Nematodes […] are" in the preceding line.

40: Likewise, replace "a plant parasite" by "plant parasites" to stay in the plural.

41: Replace "can" by "could", "may" or "might".

63: It took me a second to understand "invertebrate parasitic habits"; in this form it would mean that their habits are both parasitic and invertebrate. Please use a hyphen in "invertebrate-parasitic habits" (or reword entirely, e.g. "parasitism of invertebrates").

292: Replace "be" by "have been" simply because the mid-Cretaceous is in the past.

294: Remove the comma; it wrongly implies that H. brownii is the only mermithid you're talking about here (and would need to be matched by another comma after the name).

313-315: "depicted" refers to literal pictures; replace it by "shown", "recorded" or "demonstrated". More importantly, however, the entire clause seems like it was incompletely edited; in its current form it states that mermithids parasitized dipterans before they parasitized anything else. I think you mean: "also, the first fossil animal that was found to host a mermithid is a dipteran from Early Cretaceous Lebanese amber (Poinar et al., 1994)" or something to that effect.

316: Replace "developed" by "develop".

325: Replace "be built" by "have formed".

Figure 8: Replace "~" by "-" (Ctrl+minus on the numeric block in MS Word, Alt+0150 on the numeric block in other software); "~" will not be understood as "to" outside of China – it will be mistaken as having its mathematical meaning of "about".

*Reviewer #2 (Recommendations for the authors):*

Overall, I would highly recommend this manuscript be accepted for publication, however, I do have a few minor comments about the manuscript that I would like the authors to address before the manuscript can be accepted for publication.

L62: "their large size" should be changed to "their relatively large size" since while it is true that mermithids are fairly large in terms of nematodes, there are also many nematodes which reach much, much larger sizes than mermithids.

L175: I question naming this species after bedbugs (Cimex) since the host was a planthopper. Wouldn't it be more appropriate to name the species after the host taxa?

*Reviewer #3 (Recommendations for the authors):*

As stated before this monumental description and valuable new data on fossil parasitic parasites is timely and very welcome. I just feel you could your own observations and work a bit short by not including a supporting statistical analysis of the shift in host exploitation (at minimum an analysis with error or confidence intervals would be appropriate). You alluded to a statistical analysis in the abstract but I could not find it in the manuscript. I feel this can be easily achieved with the new literature and data you compiled and would make your interpretations much more robust and fundamental. Here are my main suggestions on how you may achieve that goal:

Line 40-41: the statements need to be backed up by more recent publications – the authors could cite De Baets et al. 2021a in this context which reviewed the nematode fossil record and the authors already cite. In addition, the authors could cite Parry et al. (2017) who most recently reported sinusoidal traces which might reflect nematode activity but also cautioned for possible alternative interpretations.

Line 59: the fossils are also treated as parasitic nematodes in another publication in the same volumes by De Baets et al. (2021b); this chapter might be of interest as the authors also depict the nematode fossil record through the Phanerozoic and clearly demonstrate that amber provides one of the richest records of nematode body fossils including parasitic ones.

Line 76-79: Indeed, also the changes in proportion for heterometabolous to holometabolous insects could be mentioned. I feel that you undersell the importance of your study by dumping this information in the supplementary material. You also allude to a statistical analysis in the abstract but as far as I can see you are just naming percentages and have pie diagrams which can be widely deceiving (e.g., Rougier et al. 2014). Depicting the changing proportions of parasitic nematodes from heterometabolous to holometabolous insects would make your study easier to follow and reproduce as well as your interpretations more convincing.

Lines 100, 114, 128, 143, 157, 174, 192, 210, 224, Below each new species, the zoobank number should be added.

Line 300: You need to back up this statement with a reference. De Baets et al. 2021 (not to be confused with the other reference you already cite) recently depicted the known record of nematode body fossils and show that amber is one of its richest sources of nematodes backing up your statement.

Line 302-305: This is an important result and therefore the information should not be buried in the supplementary material. It is crucial to analyze/depict this information in a bar plot rather than pie diagrams which although graphically appealing might distort information (compare Figure 1.3 in De Baets et al. 2021) and I feel this could be done by separating those species in heterometabolous and holometabolous insects per amber deposit. To make your interpretations more robust and deal with differences in sampling heterometabolous versus holometabolous insects, at minimum binomial error or confidence intervals should be added to this plot (Raup 1991; Takeda and Tanabe 2014) if not more broadsweeping statistical analyses. Confidence intervals are more conservative and can easily be calculated in Matlab with Binofit function or R with the binom.confint function of the package Binom as well as other software. I feel such analyses could be done for both mermithids and all (parasitic) nematodes in those 3 amber deposits. Particularly tabulated data and depiction of the distribution of mermithids in modern insects would be helpful to understand the context. This might be trivial to you but not so to the reader who does not necessarily has access to all relevant literature.

Line 330-331: In addition to De Baets et al. 2021a, 2021b would be highly relevant in this context.

Line 344-353: I appreciate this explicit ethics statement and I feel it should be mandatory in all kind of studies dealing with fossils not just amber.

Line 390: Thank you for providing the raw data but I feel your study would also benefit from depicting this data in a bar plot with confidence intervals (as explained before) as well as add data and host distributions of modern mermithids.

Suggested references:

De Baets, K., Dentzien-Dias, P., Harrison, G.W.M., Littlewood, D.T.J., Parry, L.A. (2021a). Fossil Constraints on the Timescale of Parasitic Helminth Evolution. In: De Baets, K., Huntley, J.W. (eds) The Evolution and Fossil Record of Parasitism. Topics in Geobiology 49: 231-271. Springer, Cham. (already cited)

De Baets, K., Huntley, J.W., Klompmaker, A.A., Schiffbauer, J.D., Muscente, A.D. (2021b). The Fossil Record of Parasitism: Its Extent and Taphonomic Constraints. In: De Baets, K., Huntley, J.W. (eds) The Evolution and Fossil Record of Parasitism. Topics in Geobiology 50: 1-50. Springer, Cham.

Parry, L. A., Boggiani, P. C., Condon, D. J., Garwood, R. J., Leme, J. D. M., McIlroy, D., … & Liu, A. G. (2017). Ichnological evidence for meiofaunal bilaterians from the terminal Ediacaran and earliest Cambrian of Brazil. Nature Ecology & Evolution, 1(10), 1455-1464.

Raup, D. M. (1991). The future of analytical paleobiology. Short courses in paleontology, 4, 207-216.

Rougier, N. P., Droettboom, M., & Bourne, P. E. (2014). Ten simple rules for better figures. PLoS computational biology, 10(9), e1003833.

Takeda, Y., & Tanabe, K. (2014). Low durophagous predation on Toarcian (Early Jurassic) ammonoids in the northwestern Panthalassa shelf basin. Acta Palaeontologica Polonica, 60(4), 781-794.

---

## [Author Response]

Essential revisions:1) The abstract mentions statistical analyses, but only pie charts are presented. The data you present in the supplementary material could easily, and should, be used for a statistical analysis to test your hypothesis of a shift toward holometabolous insects as hosts. Please also briefly address, in the text, the possibility of collection bias: are, for example, beetles underrepresented in museum collections of insects in amber, and are such biases likely to be identical for all sources of amber (Baltic amber has been sampled for centuries, Kachin amber only for a decade or two)?

Thanks. We fully agreed with the reviewer and revised this part. We have added a new Figure 9B and Table 1 to our paper. Indeed, collection bias is almost present in all amber biotas. However, we believe we have robust reasons to argue that the shift to holometabolous hosts does exist. Although Kachin amber has only been studied extensively in the last two decades (compared with centuries of study in Baltic amber or Dominican amber), it has become by far the most intensively studied amber biota since its Cretaceous age was appreciated, now comprising an exceptional 700 families (Ross, 2023). Also, the fossil record of holometabolous insects is clearly much better than non-holometabolous insects in Kachin amber (1296 spp. vs 465 spp. respectively). But as shown in our paper, the nematodes we found in Kachin amber are mainly associated with heterometabolous insects. Therefore, even if collection bias might exist, such as the presence of some unreported nematode-Holometabola associations, we believe our conclusion about the shift is robust. We also add some explanation in our paper, please see line 350–356 in the clean copy of our revised manuscript.

2) Please write for a slightly wider audience. For example, please explain how trophosomes are recognized, and how fossil mermithids can be differentiated from, for example, nematomorphs; please also mention if mermithids are known to exit their hosts as a panic reaction. If you can illustrate and describe the exit wounds, this would greatly strengthen the manuscript against the possibility that some of the associations might be accidental.

Thank you very much for pointing out this issue. Please see below.

3) The nomenclature needs a few minor improvements. I also urge you to take this opportunity to publish the genus name Cretacimermis properly, because that has not been done so far – the original publication lacks a diagnosis, which is required even for collective groups, so the name "Cretacimermis" is not currently available and does not even compete for homonymy.

Thanks. We formally erect this genus in this paper.

Reviewer #1 (Recommendations for the authors):I am not familiar with nematodes or amber, so I only wrote my review after receiving and reading the other reviews (I am the Reviewing Editor), and I have focused on the topics I am more competent in.Comments on scienceGiven the wide readership of eLife, it would be good to explicitly state the evidence that these rather featureless elongate fossils represent specifically the parasitic stage of, specifically, mermithids (and not for example nematomorphs or a whole new group of nematodes). In your descriptions you mention trophosomes (implying parasitism) and details of the cuticle; is any of this diagnostic for Mermithidae, or is something else that you can see? How are trophosomes recognized? (I see the brown fragments you point out in Figure 4H; why do you interpret them as a trophosome?) Is it possible to illustrate or at least describe the exit wounds in more detail in the specimens where the mermithids have fully exited?

Thank you very much for pointing out these issues. First, the parasitic stage of fossil mermithids is almost impossible to determine due to the lack of detailed structure. In fact, it is also very difficult to determine the stage of extant mermithids only based on morphological characters. It is a pity that the number of moults is based on experimental observation (Poinar, 1974), which is impossible to duplicate in fossil research.

The identification of fossil mermithids is based mainly on their relatively large size, the morphological comparison with extant mermithids (body shape, length, body diameter, tail structure, etc.), and their position and posture in relation to the insects. Other invertebrateparasitic nematodes are usually relatively short, not coiled, and much smaller than their hosts, but mermithids are relatively long, coiled, and large compared with their hosts (Poinar, 2011). Nematomorphs are also long and big, but their epicuticles are normally crossed by grooves or furrows, leaving small elevations of irregular areas (areoles) between them (Poinar, 1999, 2001, 2006). We have therefore added some explanation at the beginning of Systematic palaeontology. Please see line 86-90 in the clean copy.

Trophosome are intestines of mermithids. As mermithids develop, the intestine becomes detached from the remainder of the alimentary tract and serves as a food storage organ (trophosome) (Poinar, 2015). Trophosomes can usually be detected in fossil mermithids by their different hues (Poinar, 2011). In order to make it readable for a wider audience, we have added some arrows and explanations in our figures.

Finally, exit wounds are hard to detect even in extant insects because when mermithids have left, these wounds tend to close up. For our specimens, we indeed have evidence (direct or indirect) to argue these associations are not accidental. We revised our figures and also have added these comments as “Remarks” in Systematic palaeontology, please see lines 137–139, 154–156, 171–172, 205–207, 225–227, 245–249, 263–264, 278–279, 298 in the clean copy.

*Cretacimermis incredibilis*: As we illustrated in Figure 2D, the tail end of the mermithids is adjacent to an exit wound on the host, indicating a true parasitic association.

*Cretacimermis calypta*: Although no exit wound can be clearly found, this mermithid is closely coiled around the dragonfly, indicating that the nematode was just emerging from the host.

*Cretacimermis adelphe*: The posterior part of the abdomen of this earwig has been damaged; it is most likely that these mermithids exited from the host through this wound.

*Cretacimermis directa*: There is no distinct wound on this cricket’s body, but the nematode is adjacent to it, and there is no other insect nearby. Therefore, it is most likely that the mermithid was just emerging from the host.

*Cretacimermis longa*: These two nematodes are still in the process of exiting. Also, the abdomen of the adult cockroach is empty and therefore probably contained the developing nematode.

*Cretacimermis perforissi*: The abdomen of the first perforissid planthopper is clearly empty and probably contained the developing nematode. The abdomen of the second perforissid planthopper is broken, which could be the result of the emerging mermithid.

*Cretacimermis manicapsoci*: first piece: there is no distinct wound on the barklouse’s body, but the nematode is adjacent to it, and there is no other insect big enough nearby. Therefore, it is most likely that the mermithid was just emerging from the host. Second piece: the abdomen of the second perforissid planthopper is lost, which could be the result of the emerging mermithid.

*Cretacimermis psoci*: The abdomen of the barklouse is broken, which could be the result of the emerging of mermithid.

References:

Poinar GO, Otieno WA. 1974. Evidence of four molts in the Mermithidae. Nematologica 20:370. DOI: https://doi.org/10.1163/187529274X00456

Poinar GO, 1999. *Paleochordodes protus* n.g., n.sp. (Nematomorpha, Chordodidae), parasites of a fossil cockroach, with a critical examination of other fossil hairworms and helminths of extant cockroaches (Insecta: Blattaria). Invertebrate Biology 118:109-115. DOI: https://doi.org/10.2307/3227053

Poinar GO, 2001. Nematoda and Nematomorpha, in: Thorp JH, Covich AP (Eds.), Ecology and classification of North American freshwater invertebrates, Second ed. Academic Press, New York, pp. 255–295.

Poinar GO, Buckley R. 2006. Nematode (Nematoda: Mermithidae) and hairworm (Nematomorpha: Chordodidae) parasites in Early Cretaceous amber. Journal of Invertebrate Pathology 93:36–41. DOI: https://doi.org/10.1016/j.jip.2006.04.006

Poinar GO, 2011. The evolutionary history of nematodes: as revealed in stone, amber and mummies. Brill, Amersfoort, the Netherlands.

Poinar GO, 2015. Chapter 14 – Phylum Nemata, in: Thorp JH, Rogers DC (Eds.), Thorp and Covich's Freshwater invertebrates (Fourth Edition). Academic Press, Boston, pp. 273–302.

Can you exclude collection bias as a factor in the apparent shift to holometabolous hosts?

Thanks. The collection bias does not affect our result. Please see Essential revision #2.

Line 25: By "statistical analyses", do you mean the pie charts in Figure 9? That's a presentation of data, not an analysis. The term made me expect mathematics.

Thanks. We calculated the 95% CI using the Agresti-Coull method of the “binom.confint” function from the binom *R* package (https://cran.r-project.org/package=binom) of *R* 4.2.2. We also added a new Figure 9B and Table 1 to our paper.

64: Are mermithids known to exit their hosts as a flight reaction when the host is stressed?

Thank you. Yes, nematodes tend to leave their hosts when their hosts are threated, e.g., caught by resin flow. That is why many fossil nematodes that occur in amber are very close to their hosts or just emerging from their hosts. We also add some words to explain this. Please see lines 53–56 in the clean copy.

Comments on nomenclatureI was wrong in my pre-review: collective groups are explicitly allowed under the ICZN; and diagnoses do not need to be limited to morphological characters, they can refer to behavior. However, collective species are not allowed – only collective genera and subgenera (ICZN Articles 10.3, 42.2.1). The species names you propose are therefore fine, just don't call them "collective".

Thanks. We deleted all the “Collective species” before our new species.

However, because genus names must be published with a diagnosis even if they're intended for a collective genus, "Cretacimermis" is actually unavailable: for purposes of zoological nomenclature, this name does not exist. I quote the original publication (Poinar 2001b: 262): "Since the generic placement of this specimen was based in part on its host family, it is now prudent to transfer the species libani from Heleidomermis Rubstov into a new genus Cretacimermis Poinar. Thus, this nematode, which is the oldest definite fossil nematode, should now be called Cretacimermis libani Poinar, Acra and Acra (1994)." This is not "a description or definition that states in words characters that are purported to differentiate the taxon" (ICZN Art. 3.1.1). I therefore strongly recommend that you validly publish Cretacimermis here as a new name. (Interestingly, you don't need to select a type species, because collective groups do not have type species: ICZN Art. 13.3.2, 42.3.1, 66, 67.14. This must be precisely because species cannot be collective groups.)

Thank you very much for pointing out this issue. We formally erect this genus in this paper. Collectives do not contain type species because they are not natural genera. Please see lines 104–123 in the clean copy.

I also strongly recommend against using times or places in diagnoses (if I find, say, a mermithid exiting an archaeognath in Late Cretaceous amber from New Jersey, do I really have to erect a new species name for it instead of referring it to "Cretacimermis" incredibilis…?), let alone the type of preservation (amber), but this is not forbidden by the ICZN. ("Character", as used in Art. 3.1.1, is defined in the Glossary as: "Any attribute of organisms used for recognizing, differentiating, or classifying taxa.")

Thanks. We also do not want to simply use times or places in diagnoses; however, it is almost impossible to identify fossil nematodes at generic and/or specific level based on morphology and we therefore have to use some other information in diagnoses like times, places and hosts as valuable circumstantial evidence. The same method is widely used in fossil nematodes. For example, *Heydenius* Taylor, 1935 was erected as a collective genus for all fossil mermithids from the Cenozoic, and *Heydenius neotropicus* Poinar, 2011 is for those infecting Chironomidae from Dominican amber, *Heydenius matutinus* (Menge, 1866) Taylor, 1935 is for those infecting Chironomidae from Baltic amber. Thus, if we do find a mermithid exiting an archaeognathan in Late Cretaceous amber from New Jersey, a new species is reasonable as unlikely to have infected chironomids in Dominican amber. It is therefore a more natural solution.

Line 101: Why "extraordinary" and not simply "incredible"?

Thanks. Corrected.

129: The species name itself is fine, but adelphos means "brother", not "sister"; "sister" is "adelphê".

Thanks. Corrected. We are very grateful for your comments and suggestions about the nomenclature.

144: Much simpler: "The species epithet is the Latin 'directa' = arranged in a straight line."

Thanks. Corrected.

172, 175, 193, 225: Another reviewer has pointed out that not all bugs are bedbugs, and that naming a planthopper parasite after bedbugs may therefore not be a good idea. Given that you name "C." manicapsoci after the type genus of Manicapsocidae (not directly after Manicapsocidae) and likewise "C." cecidomyiae after they type genus of Cecidomyiidae, you could also name the planthopper parasite "C. perforissi", after the type genus of Perforissidae.

Thanks. Corrected.

226: Being a valid name, Cecidomyia should be preceded by a comma rather than surrounded by quotation marks. You can also delete it entirely; you didn't spell out Manicapsocus in line 193.

Thanks. Deleted.

286: Replace "Archeognatha" by "Archaeognatha".

Thanks. Corrected.

365: Is he Peter or Petr?

Thanks. He is Peter

593-594 and Supplementary Table: As you correctly state in the text, the authors of "C." chironomae are not Poinar, 2011, but Grimaldi et al., 2002.

Thanks. However, although Grimaldi et al., 2002 first reported two mermithid nematode parasites emerging from the abdomen of a female chironomid midge, they did not describe it. It was “Poinar, 2011” who formally described these nematodes and proposed this new species.

Comments on style and languageLine 27-28: "have […] exploited" is from the point of view of the present, so it clashes with "until the Cenozoic"; replace it by "had […] exploited".

Thanks. Corrected.

34: Replace "one" by "some" to fit the plural "Nematodes […] are" in the preceding line.

Thanks. Corrected.

40: Likewise, replace "a plant parasite" by "plant parasites" to stay in the plural.

Thanks. We replaced “fossils” by “fossil” since there is only one nematode known from Rhynie Chert so far.

41: Replace "can" by "could", "may" or "might".

Thanks. Corrected.

63: It took me a second to understand "invertebrate parasitic habits"; in this form it would mean that their habits are both parasitic and invertebrate. Please use a hyphen in "invertebrate-parasitic habits" (or reword entirely, e.g. "parasitism of invertebrates").

Thanks. Corrected.

292: Replace "be" by "have been" simply because the mid-Cretaceous is in the past.

Thanks. Corrected.

294: Remove the comma; it wrongly implies that H. brownii is the only mermithid you're talking about here (and would need to be matched by another comma after the name).

Thanks. Corrected.

313-315: "depicted" refers to literal pictures; replace it by "shown", "recorded" or "demonstrated". More importantly, however, the entire clause seems like it was incompletely edited; in its current form it states that mermithids parasitized dipterans before they parasitized anything else. I think you mean: "also, the first fossil animal that was found to host a mermithid is a dipteran from Early Cretaceous Lebanese amber (Poinar et al., 1994)" or something to that effect.

Thanks. Corrected.

316: Replace "developed" by "develop".

Thanks. Corrected.

325: Replace "be built" by "have formed".

Thanks. Corrected.

Figure 8: Replace "~" by "-" (Ctrl+minus on the numeric block in MS Word, Alt+0150 on the numeric block in other software); "~" will not be understood as "to" outside of China – it will be mistaken as having its mathematical meaning of "about".

Thanks. Corrected.

Reviewer #2 (Recommendations for the authors):Overall, I would highly recommend this manuscript be accepted for publication, however, I do have a few minor comments about the manuscript that I would like the authors to address before the manuscript can be accepted for publication.L62: "their large size" should be changed to "their relatively large size" since while it is true that mermithids are fairly large in terms of nematodes, there are also many nematodes which reach much, much larger sizes than mermithids.

Thanks. Corrected.

L175: I question naming this species after bedbugs (Cimex) since the host was a planthopper. Wouldn't it be more appropriate to name the species after the host taxa?

Thanks. We renamed this species as *Cretacimermis perforissi* based on the editor’s comments (please see comment 16).

Reviewer #3 (Recommendations for the authors):As stated before this monumental description and valuable new data on fossil parasitic parasites is timely and very welcome. I just feel you could your own observations and work a bit short by not including a supporting statistical analysis of the shift in host exploitation (at minimum an analysis with error or confidence intervals would be appropriate). You alluded to a statistical analysis in the abstract but I could not find it in the manuscript. I feel this can be easily achieved with the new literature and data you compiled and would make your interpretations much more robust and fundamental. Here are my main suggestions on how you may achieve that goal:

Thanks. We have added a new Figure 9B and Table 1 to our paper.

Line 40-41: the statements need to be backed up by more recent publications – the authors could cite De Baets et al. 2021a in this context which reviewed the nematode fossil record and the authors already cite. In addition, the authors could cite Parry et al. (2017) who most recently reported sinusoidal traces which might reflect nematode activity but also cautioned for possible alternative interpretations.

Thanks. We have added these two references here.

Line 59: the fossils are also treated as parasitic nematodes in another publication in the same volumes by De Baets et al. (2021b); this chapter might be of interest as the authors also depict the nematode fossil record through the Phanerozoic and clearly demonstrate that amber provides one of the richest records of nematode body fossils including parasitic ones.

Thanks. We added this reference here.

Line 76-79: Indeed, also the changes in proportion for heterometabolous to holometabolous insects could be mentioned. I feel that you undersell the importance of your study by dumping this information in the supplementary material. You also allude to a statistical analysis in the abstract but as far as I can see you are just naming percentages and have pie diagrams which can be widely deceiving (e.g., Rougier et al. 2014). Depicting the changing proportions of parasitic nematodes from heterometabolous to holometabolous insects would make your study easier to follow and reproduce as well as your interpretations more convincing.

Thanks. We have added a new Figure 9B and Table 1 to our paper.

Lines 100, 114, 128, 143, 157, 174, 192, 210, 224, Below each new species, the zoobank number should be added.

Thanks. Added.

Line 300: You need to back up this statement with a reference. De Baets et al. 2021 (not to be confused with the other reference you already cite) recently depicted the known record of nematode body fossils and show that amber is one of its richest sources of nematodes backing up your statement.

Thanks. This reference is informative and we added this reference.

Line 302-305: This is an important result and therefore the information should not be buried in the supplementary material. It is crucial to analyze/depict this information in a bar plot rather than pie diagrams which although graphically appealing might distort information (compare Figure 1.3 in De Baets et al. 2021) and I feel this could be done by separating those species in heterometabolous and holometabolous insects per amber deposit. To make your interpretations more robust and deal with differences in sampling heterometabolous versus holometabolous insects, at minimum binomial error or confidence intervals should be added to this plot (Raup 1991; Takeda and Tanabe 2014) if not more broadsweeping statistical analyses. Confidence intervals are more conservative and can easily be calculated in Matlab with Binofit function or R with the binom.confint function of the package Binom as well as other software. I feel such analyses could be done for both mermithids and all (parasitic) nematodes in those 3 amber deposits. Particularly tabulated data and depiction of the distribution of mermithids in modern insects would be helpful to understand the context. This might be trivial to you but not so to the reader who does not necessarily has access to all relevant literature.

Thank you very much for pointing out these issues. We also realize this drawback and we have now calculated the 95% CI using the Agresti-Coull method of the “binom.confint” function from the binom *R* package (https://cran.r-project.org/package=binom) of *R* 4.2.2. We also added a new Figure 9B and Table 1 in our paper to address this problem. We did not prepare a similar figure for mermithids alone because the dataset is too small at present. Meanwhile, since we compiled the “occurrence” of invertebrate–nematode associations from these amber localities, it is impossible to compare with modern mermithids. For example, the parasite of *Cretacimermis chironomae* occurs five times in Kachin amber, but an extant dipteran-parasitized mermithid species can occur many times just in a single pond. However, it is evident that mermithids as well as all invertebrate-parasitized nematodes prefer to infect holometabolous insects rather than other invertebrates (Poinar, 1975; Poinar, personal observation). We have also added some explanations in our paper, please see lines 367–369 in the clean copy.

Line 330-331: In addition to De Baets et al. 2021a, 2021b would be highly relevant in this context.

Thanks. We have added this reference here.

Line 344-353: I appreciate this explicit ethics statement and I feel it should be mandatory in all kind of studies dealing with fossils not just amber.

Thanks for your positive comment.

Line 390: Thank you for providing the raw data but I feel your study would also benefit from depicting this data in a bar plot with confidence intervals (as explained before) as well as add data and host distributions of modern mermithids.

Thanks. We have added a new Figure 9B and Table 1 to our paper.